# Achieving Subcategorical Erasure in Text-to-Image Models

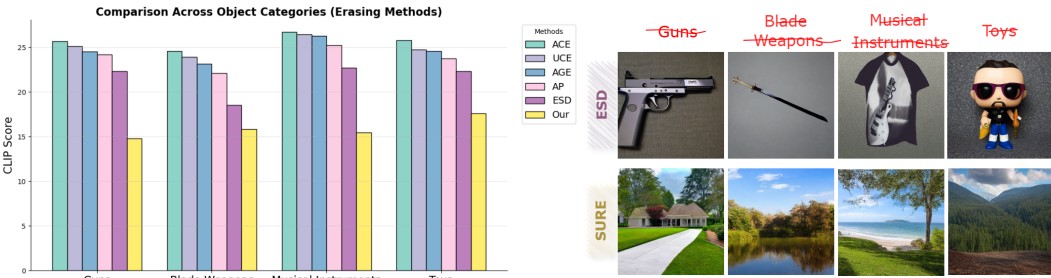

Figure 1: The figure illustrates similarity scores (left) for different object categories across various erasure methods, where each category is used as the erasure target. Lower scores correspond to better erasure. Existing methods, including ESD, ACE, UCE, AGE, and AP, often fail to erase sub-categories within a given category completely. For example, in the case of the guns category, ESD fails to remove sub-categories such as pistols fully, as shown in the generated samples on the right, whereas our approach (SURE) successfully achieves sub-categorical level erasure. The qualitative results (right) further show that our approach (SURE) eliminates targeted categories (e.g., guns, blade weapons, musical instruments, and toys) completely.

## Abstract

The emergence of large-scale text-to-image diffusion (T2ID) models has led to significant advancements in generating high-quality visual content from textual prompts. However, these powerful capabilities have also raised growing concerns about the generation of harmful and copyrighted material. While existing concept erasure techniques can effectively block the production of specific unwanted concepts from prompts, they often fall short when it comes to erasing an entire category (including subcategories) and are typically limited to handling only a few concepts at a time. In this paper, we introduce **S**ubcategorical **U**nlearning via **R**egularized **E**rasure (SURE), a novel method for removing entire subcategories from text-to-image diffusion models using only a single parent category as the target. Unlike prior approaches, SURE does not rely on sets of synonyms. Instead, it employs concept space to discover and eliminate the target category while preserving the model's overall utility. To further enhance erasure, SURE incorporates Lipschitz regularization, which encourages smoother model responses to perturbations around the target category. Specifically, the regularization promotes consistent behavior in the model's latent space when exposed to slight variations of the category to be forgotten. This smoothness constraint aids in erasure while maintaining the model's ability to generate unrelated content. Extensive experiments conducted across three tasks—object removal, suppression of explicit content, and elimination of artistic styles demonstrate that SURE achieves balanced performance in both effective category erasure and preservation of non-target concepts.

## 1 Introduction

Text-to-image generative models (Ding et al., 2022; Chang et al., 2023; Ramesh et al., 2022; Nichol et al., 2021; Lu et al., 2023) have made significant advancements in generating high-quality im-

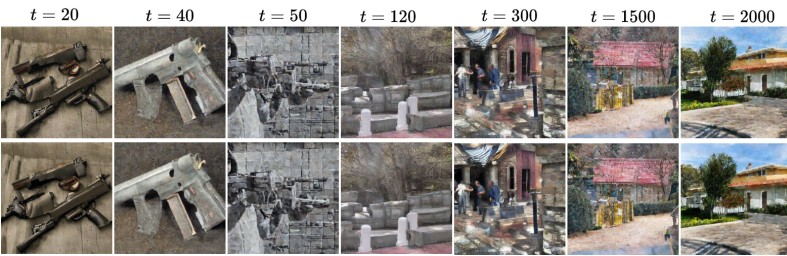

Figure 2: Visual illustration of perturbed (top row) and original (bottom row) image samples used during the computation of the Lipschitz regularization in the context of the gun category erasure at different training steps. Perturbations are applied to the original inputs to encourage smooth behavior in the model's latent space around the gun category.

ages almost instantly from simple textual prompts. These models learn to replicate diverse visual concepts from large-scale datasets (Carlini et al., 2023) of image-caption pairs from the internet. However, this extensive model training comes with considerable risks, including the generation of copyrighted material (Jiang et al., 2023), privacy-invading content (Somepalli et al., 2023), and explicit imagery (Schramowski et al., 2023), all of which raise safety concerns.

To address the associated risks, initial efforts primarily focused on dataset filtering during training (StabilityAI, 2022; Carlini et al., 2022), post-generation filtering (Rando et al., 2022), and inference-time guidance (Schramowski et al., 2023). Dataset filtering removes undesirable content but requires significant human effort and expensive retraining. Post-generation filtering (SmithMano, 2022) is prone to false positives and can be easily bypassed by users, whereas inference-time guidance provides a lightweight alternative but remains vulnerable to adversarial attacks (Yang et al., 2024). An emerging alternative is *concept erasure* (Bui et al., 2024; Lu et al., 2024; Gandikota et al., 2023; Bui et al., 2025; Wu et al., 2024), which involves fine-tuning a pre-trained model to unlearn specific concepts described in a text prompt (Huang et al., 2024), preventing the generation of related image content without requiring full retraining. There are primarily two types of fine-tuning methods for concept erasure: Attention-based methods (Huang et al., 2024; Zhang et al., 2023; Orgad et al., 2023; Gandikota et al., 2024; Lu et al., 2024) and Output-based methods (Gandikota et al., 2023; Wu et al., 2024; Bui et al., 2025; 2024). Attention-based methods focus on modifying the attention mechanisms within models to eliminate undesirable concepts. In contrast, Output-based methods aim to optimize the output image by minimizing the difference between the predicted noise and the target noise.

While prior methods have demonstrated success in removing specific undesirable concepts, most are limited to erasing either a single concept (e.g., pistols) or a small set of closely related concepts (e.g., revolvers, rifles, and shotguns) at a time. However, these approaches often struggle to eliminate complete category (e.g., guns), including all subcategories, effectively. As a result, unwanted content may still be generated by the model (ref. Fig. 1). For instance, when erasing the category of guns, the model must automatically erase subcategories such as SMG, revolvers, pistols, rifles, and shotguns. While eliminating specific harmful content is important, it is equally critical to enable the removal of entire category (including all subcategories) to improve the safety and general applicability of text-to-image models. Some methods (Lu et al., 2024; Li et al., 2025) require individual concepts for multi-concept erasure, which may not be practical, as not all subcategories are known and their number can be very large, making it computational expensive. Instead, we propose erasing entire subcategories from text-to-image diffusion models using only a single parent category as the target. We follow the WordNet 3.1 (Miller, 1995) ontology definitions of hypernymy and hyponymy to distinguish between categories (hypernyms) and subcategories (hyponyms) (ref. Appendix A.2 for more details). In this context, a category refers to a grouping of subcategories. For example, "flower" is a category that encompasses multiple subcategories such as rose, tulip, and daisy.

To this end, we introduce **S**ubcategorical **U**nlearning via **R**egularized **E**rasure (SURE), a novel framework for removing entire subcategories from text-to-image diffusion models using only a single parent category prompt. SURE ensures the automatic elimination of multiple subcategories (e.g., SMG, revolvers, pistols, rifles, and shotguns) within a category (e.g., Guns) by erasing a single parent category (e.g., Guns). Moreover, SURE not only removes range of subcategories under the targeted category but also achieves a strong balance between Efficacy (the model erases only

the intended target category) and Locality (the model retains its ability to generalize to unrelated categories). Our core design builds upon extending Lipschitz continuity (Yoshida & Miyato, 2017) to diffusion-based text-to-image models for robust unlearning. We adapt Lipschitz regularization directly to the latent space of generated image samples, encouraging smooth generative behavior around the target category. Specifically, we generate several perturbed versions (see Fig. 2) of the given target category and train the model to minimize the difference between the original output and the outputs of these perturbed categories. This process smooths the model in targeted regions (Foster et al., 2024; Kravets & Namboodiri, 2025), leading to the erasure of the target category (where the target region refers to an unseen concept space), while preserving the model's generalization performance. Furthermore, we incorporate a neutral eraser and a preservation objective, which guide the model in retain its behavior on neutral concepts while effectively erasing the undesirable subcategories (ref. Sec. 4.1). We extensively evaluate the proposed approach across three erasure tasks: object category removal, explicit content suppression, and artistic style elimination. Our method demonstrates strong performance in erasing entire subcategories. It achieves a better balance between efficacy and locality compared to state-of-the-art (SOTA) methods.

The contributions of this work are summarized as follows: We propose SURE, a novel framework for entire subcategories erasure in pre-trained text-to-image (T2I) models. Unlike prior approaches that focus on individual concept or small concept sets, SURE removes entire subcategories of undesirable content using only a single parent category prompt. SURE employs Lipschitz regularization, along with a neutral eraser and a preservation objective, to strike a balance between effective forgetting of the target category and retention of unrelated, desirable content. Our joint formulation introduces a scalable approach to category Erasure in text-to-image models. We validate this framework across diverse tasks, including object category removal, artistic style elimination, and explicit content erasure, showing consistent improvements over state-of-the-art methods.

## 2 Related Works

**Fine-tuning-based Concept Erasing:** These methods (Gao et al., 2025; Meng et al., 2024; Yao et al.) adapt pre-trained foundation models to eliminate specific target concepts by modifying their parameters, offering a scalable alternative to retraining from scratch. One class of these methods focuses on attention-based modifications. For example, Forget-Me-Not (Zhang et al., 2023) employs attention re-steering by fine-tuning only the UNet to suppress intermediate attention maps related to the target concept. TIME (Orgad et al., 2023) adjusts the linear projections for keys and values within cross-attention layers to redirect the influence of the unwanted concept toward a more neutral interpretation. MACE (Lu et al., 2024) applies a closed-form cross-attention refinement using multiple LoRA modules for efficient fine-tuning. MACE evaluates generality by using synonyms of the concepts, but this does not guarantee the evaluation of all subcategories. UCE (Gandikota et al., 2024) proposes a closed-form erasure solution while preserving non-target concepts through an explicit preservation objective. AdaVD (Wang et al., 2024) calculates the orthogonal complement of each cross-attention and uses the shift factor to adaptively adjust the erasure strength, enhancing effective prior preservation without compromising erasure efficacy. Another category of fine-tuning-based approaches directly optimizes model outputs to ensure the removal of unwanted content. ESD (Gandikota et al., 2023) maps the distribution of a target concept to that of a neutral or "null" prompt, effectively disassociating it from the generated image. Methods like, AGE (Bui et al., 2025), and AP (Bui et al., 2024) take a similar approach by mapping the target concept to either a neutral, null, or adversarial concept. Anti-Editing Concept Erasure (Wang et al., 2025) not only erases the target concept during generation but also filters it out during editing.

**Adversarially Robust Concept Erasing:** In the context of adversarial attacks, adversarial examples are carefully crafted inputs designed to manipulate models into producing unintended or harmful outputs. Recently, several methods have focused on enhancing the robustness of concept erasure in fine-tuned models. AdvUnlearn (Zhang et al., 2024a) improve robustness by alternately training the model for both concept erasure and adversarial resistance. Specifically, AdvUnlearn (Zhang et al., 2024a) incorporates additional regularization to balance the removal of target concepts while preserving unrelated ones. UnlearnDiff (Zhang et al., 2024b) addresses optimization challenges in adversarial settings using Projected Gradient Descent to ensure effective erasure. RACE. (Kim et al., 2024) identifies adversarial text embeddings that can potentially reconstruct erased concepts and targets them for removal. Similarly, methods like AP (Bui et al., 2024) and AGE (Bui et al., 2025) identify sets of concepts most affected by parameter changes—termed adversarial concepts—and use

these to guide stable and robust concept erasure while preserving unrelated generation capabilities. In STEREO author (Srivatsan et al., 2024) adopts a two-stage strategy involving explicit min-max optimization: the first stage employs adversarial training, while the second uses an anchor-concept-based compositional objective to erase the target concept in a single fine-tuning pass. Most existing methods (ref. Fig. 1) fall short in achieving complete category (including all subcategories) erasure. Addressing this gap, our work introduces an output-based fine-tuning approach aimed at removing entire subcategories, marking an advancement in strengthening text-to-image (T2I) diffusion models against subcategory-level undesirable concepts.

## 3 PRELIMINARIES

### 3.1 DENOISING DIFFUSION MODELS

Diffusion models (Ho et al., 2020) represent a class of generative models capable of synthesizing high-quality, realistic images. These models consist of two main stages: the forward diffusion process, in which noise is incrementally introduced to an input image, and the reverse denoising process, where the model attempts to predict the noise component $\epsilon_t$ added in the forward process. Formally, given a sequence of $T$ diffusion steps $x_0, x_1, \ldots, x_T$, the denoising process can be expressed as:

$$p_\theta(x_{T:0}) = p(x_T) \prod_{t=1}^{T} p_\theta(x_{t-1}|x_t), \tag{1}$$

where the model is optimized by minimizing the difference between the actual noise $\epsilon$ and the noise estimate $\epsilon_\theta(x_t, t)$ provided by the denoising model $\theta$ at step $t$, as follows:

$$\mathcal{L} = \mathbb{E}_{x_0 \sim p_{\text{data}}, t, \epsilon \sim \mathcal{N}(0, I)} \left[ \|\epsilon - \epsilon_\theta(x_t, t)\|_2^2 \right]. \tag{2}$$

### 3.2 LATENT DIFFUSION MODELS

Latent diffusion models (LDMs) (Rombach et al., 2022) enhance the efficiency of the diffusion process by operating within a lower-dimensional latent space $z$, obtained via a pretrained variational autoencoder (VAE) comprising an encoder $f_\theta(\cdot)$ and a decoder $D$. During training, an image $x$ is mapped to its latent representation $z = f_\theta(x)$, where noise is progressively added, yielding $z_t$ with increasing noise levels over time. LDMs function as a sequence of denoising models, parameterized by $\theta$, that estimate the noise component $\epsilon_\theta(z_t, c, t)$. This estimation is conditioned on both the timestep $t$ and an auxiliary conditioning variable $c$, which corresponds to the CLIP embedding of an input prompt. The optimization objective of LDMs is given by:

$$L = \mathbb{E}_{z_t \in \mathcal{E}(x), t, c, \epsilon \sim \mathcal{N}(0, 1)} \left[ \|\epsilon - \epsilon_\theta(z_t, c, t)\|_2^2 \right]. \tag{3}$$

To enhance and control image generation, classifier-free guidance (Ho & Salimans, 2022) is employed. This technique adjusts the probability distribution in favor of samples that align with an implicit classifier $p(c|z_t)$. During inference, the model is trained on both conditional and unconditional denoising, allowing the computation of both conditional and unconditional noise estimates. The final guided noise estimate $\tilde{\epsilon}_\theta(z_t, c, t)$ is computed as:

$$\tilde{\epsilon}_\theta(z_t, c, t) = \epsilon_\theta(z_t, t) + \alpha(\epsilon_\theta(z_t, c, t) - \epsilon_\theta(z_t, t)), \tag{4}$$

where $\alpha > 1$ is the guidance scale controlling the strength of conditioning. The inference process initiates with a Gaussian noise sample $z_T \sim \mathcal{N}(0, 1)$. Using the refined noise estimate $\tilde{\epsilon}_\theta(z_T, c, T)$, the model iteratively denoises $z_T$ to obtain $z_{T-1}$. This iterative refinement continues until $z_0$ is reached, which is then transformed back to image space using the decoder $x_0 \leftarrow D(z_0)$.

## 4 METHOD

We aim to develop a framework that removes multiple subcategories using a single parent category from pretrained T2I diffusion models. To achieve this, we will fine-tune a new eraser model, $\theta'$, (ref. Fig. 3) with the objective of unlearning the target parent category while preserving the model's generalization to unrelated categories. We also identify related subcategories, such as synonyms, and enable the eraser model to eliminate these related subcategories as well.

### 4.1 CONCEPT ERASING WITH A NEUTRAL ERASER AND PRESERVATION

The objective of concept erasure is to remove an undesirable concept $c_e \in \mathbf{E}$, where $\mathbf{E}$ represents the set of concepts to be erased. $c_e$ is a category. Given an associated textual description, we typically

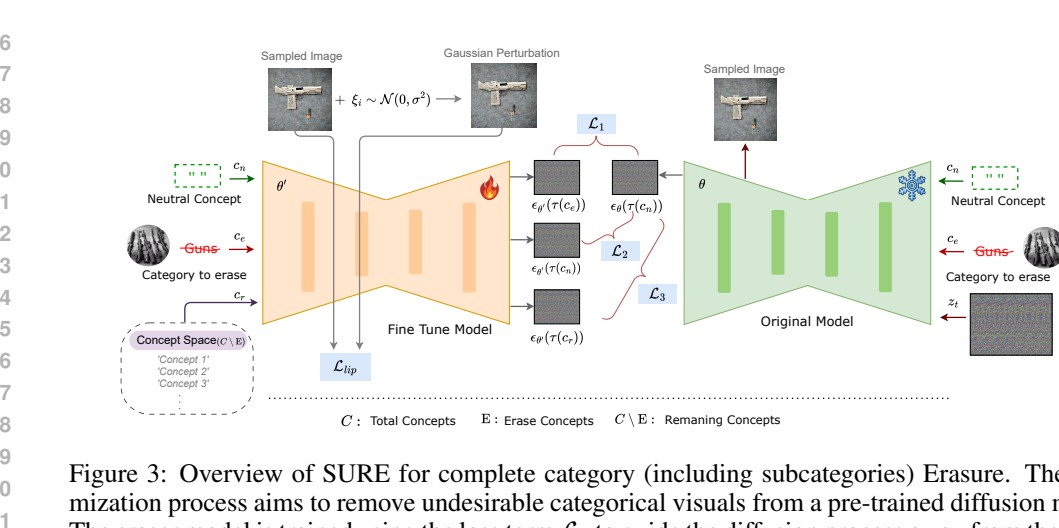

Figure 3: Overview of SURE for complete category (including subcategories) Erasure. The optimization process aims to remove undesirable categorical visuals from a pre-trained diffusion model. The eraser model is trained using the loss term $\mathcal{L}_1$ to guide the diffusion process away from the target category. To preserve unrelated content, $\mathcal{L}_2$ regularizes the model's outputs to remain consistent for a neutral concept. Related subcategories are selected from the remaining concept space and mapped to the neutral concept, enabling effective erasure via $\mathcal{L}_3$. Additionally, Lipschitz regularization $\mathcal{L}_{lip}$ smooths the model's behavior with respect to input perturbations, facilitating stable erasure of the target category while maintaining the model's performance on non-target categories.

fine-tune a pretrained text-to-image diffusion model $\epsilon_\theta(z_t, \tau(c), t)$, resulting in an updated model $\epsilon_{\theta'}$ with parameters $\theta'$, which produces an output $\epsilon_{\theta'}(z_t, \tau(c), t)$.

For simplicity, we represent the outputs as $\epsilon_\theta(\tau(c))$ and $\epsilon_{\theta'}(\tau(c))$. To erase specific visual category $c_e$ from a pre-trained diffusion model, we adjust the model's predictions away from the erased category. This is achieved by mapping the erased category to a neutral concept (Gandikota et al., 2023; Orgad et al., 2023). We use empty string " " as a neutral concept in our implementation. The objective is defined as:

$$\min_{\theta'} \underbrace{\mathbb{E}_{c_e \in \mathbf{E}} \left[ \| \epsilon_{\theta'}(\tau(c_e)) - \epsilon_\theta(\tau(c_n)) \|_2^2 \right]}_{\mathcal{L}_1} \tag{5}$$

From the objective function (ref. Eq. 5), the eraser model is trained to decrease the likelihood that the generated image contains undesirable category visuals by minimizing the L2 distance between $\epsilon_{\theta'}$ and $\epsilon_\theta$, thus successfully removing the category. The above objective effectively removes the undesirable category; however, it negatively impacts (Bui et al., 2025; Gandikota et al., 2024) the model's ability to preserve other unrelated categories.

To retain unrelated concepts during concept erasure, prior methods (Lu et al., 2024; Gandikota et al., 2024) explicitly preserve large sets of unrelated image–text pairs. However, this strategy is not practical when dealing with category erasure. For instance, when erasing the concept of "guns", it is infeasible to manually enumerate and preserve hundreds of unrelated categories (e.g., horses, microwaves, scissors, etc.). ACE (Wang et al., 2025) also highlights that traversing preserved concepts is challenging, as pre-trained models cover a vast general semantic space. To ensure the retention of unrelated categories, SURE utilize a regularization constraint (see Eq. 6) that encourages the model to maintain its behavior on neutral concepts while erasing undesirable ones (Bui et al., 2025). Specifically, we add a regularization loss to ensure that the model's predictions for the neutral concept remain consistent before and after fine-tuning. This is achieved by minimizing the difference between the outputs of the fine-tuned and original models for the neutral concept ($c_n$). By anchoring the model at a known neutral point, the fine-tuning process is constrained from drifting too far from the original distribution, thereby helping preserve the model's general generation capabilities. The regularization term can be formulated as follows:

$$\underbrace{\| \epsilon_{\theta'}(\tau(c_n)) - \epsilon_\theta(\tau(c_n)) \|_2^2}_{\mathcal{L}_2} \tag{6}$$

## 4.2 RELATED SUB CATEGORIES ERASURE

Related subcategories are the acronyms for the subcategories that are closely related to the category. We use adversarial learning (Bui et al., 2025) to identify related sub categories within the concept space $C$. To this end, we optimize the following objective by maximizing $\mathcal{L}_3$ (ref. Eq. 7) with respect to $c_r$. In other words, it seeks a concept $c_r$ whose output deviates the most from the neutral concept $c_n$. Since this is a maximization objective, we want the related sub categories to remain distinct from the null concept. In essence, the goal is to "maximize the differences between all related sub categories and the null concept after erasure." The maximization objective is given below:

$$\max_{c_r \in C \backslash \mathrm{E}} \underbrace{\|\epsilon_{\theta'}(\tau(c_r)) - \epsilon_\theta(\tau(c_n))\|_2^2}_{\mathcal{L}_3} \tag{7}$$

Here, the related sub categories is selected from the remaining concepts in the concept space, i.e., $C \backslash \mathrm{E}$, which represents the set difference between $C$ and $E$. The equation above enforces the mapping of the related sub categories to the neutral concept, facilitating the erasure of the related sub categories with respect to the $\theta'$. More details are provided in the appendix A.4.

## 4.3 LIPSCHITZ REGULARIZATION FOR ERASURE

Given the category we wish to erase, we define an optimization problem to reduce the trained model's sensitivity to this category. We aim to achieve this by 'smoothing' the model's responses to the category concerning perturbations in the input image. By training the model to align its output for a target concept with its random perturbations, we can successfully reduce memorization of that target category (i.e., forget). This also links with Feldman (2020) Feldman (2020), which observes that generalized models are often forced to memorize certain information; our objective is to protect the broader generalization while removing the relevant memorized data. Importantly, Lipschitz regularization reduces the model's sensitivity to small perturbations of concept variants. In category erasure, this is crucial, as the model must forget not only the explicit category prompt (e.g., "gun") but also subcategorical variants (e.g., "rifle," "SMG," "firearm").

We extend Lipschitz regularization (Foster et al., 2024) to be applied to intermediate image outputs generated by the erasure model to enforce smoothness in the latent space representation for small input perturbations. Let $x$ be an image generated by the model conditioned on a category embedding $c_e$. We apply a small perturbation $\xi$ to $x$, where $\xi$ is independently sampled from a Gaussian distribution, i.e., $\xi \sim \mathcal{N}(0, \sigma^2)$. This results in a perturbed image $x' = x + \xi$. Both $x$ and $x'$ are passed through the first-stage VAE encoder $f_{\theta'}(\cdot)$ to obtain their latent embeddings. The Lipschitz regularization loss is defined as follows:

$$\mathcal{L}_{lip} = \mathbb{E}\left(\frac{\|f_{\theta'}(x) - f_{\theta'}(x+\xi)\|_2}{\|x - (x+\xi)\|_2}\right) \approx \frac{1}{N}\sum_{i=1}^{N}\left(\frac{\|f_{\theta'}(x) - f_{\theta'}(x+\xi_i)\|_2}{\|\xi_i\|_2}\right), \tag{8}$$

where $\xi_i$ is the $i$-th perturbation sample and $N$ is the total number of perturbation samples. We do not compute gradients with respect to $f_{\theta'}(.)$ during optimization, instead these are treated as fixed. Further details are provided in the appendix A.3.

## 4.4 FINAL ERASURE OBJECTIVE

Our final objective function is a combination of all four objectives: $\mathcal{L}_1$, $\mathcal{L}_2$, $\mathcal{L}_3$ and $\mathcal{L}_{lip}$. Each loss term plays a crucial role in category erasure from the pre-trained model, with specific functionalities. The overall objective is as follows:

$$\min_{\theta'} \underbrace{\mathbb{E}_{c_e \in \mathbf{E}} \|\epsilon_{\theta'}(\tau(c_e)) - \epsilon_\theta(\tau(c_n))\|_2^2}_{\mathcal{L}_1} + \underbrace{\|\epsilon_{\theta'}(\tau(c_n)) - \epsilon_\theta(\tau(c_n))\|_2^2}_{\mathcal{L}_2}$$

$$+ \max_{c_r \in C \backslash \mathrm{E}} \underbrace{\|\epsilon_{\theta'}(\tau(c_r)) - \epsilon_\theta(\tau(c_n))\|_2^2}_{\mathcal{L}_3} + \underbrace{\mathbb{E}\left(\frac{\|f_{\theta'}(x) - f_{\theta'}(x+\xi)\|_2}{\|x - (x+\xi)\|_2}\right)}_{\mathcal{L}_{lip}} \tag{9}$$

We minimize the objective $\mathcal{L}_1$ with respect to $\theta'$, ensuring that the distance between the erased model's output for the erased category and the original model's output for the neutral concept is minimized. This facilitates the erasure of the undesirable category. Similarly, the regularization

Table 1: Comparison of concept erasure methods across four categories: Guns, Bladed Weapons, Musical Instruments, and Toys. We report the following metrics: $Acc_E$ — accuracy on erased concepts ($\downarrow$), $Acc_L$ — locality accuracy on non-erased (remaining) concepts ($\uparrow$), and $H$ — harmonic mean ($\uparrow$). Notably, SURE achieves the highest harmonic mean across all four categories, demonstrating its effectiveness in erasing entire categories (including subcategories) using only the parent category label.

| Method | Venue | Guns | | | Blade Weapons | | | Musical Instruments | | | Toys | | |
|--------|-------|------|------|------|------|------|------|------|------|------|------|------|------|
| | | $Acc_E \downarrow$ | $Acc_L \uparrow$ | $H \uparrow$ | $Acc_E \downarrow$ | $Acc_L \uparrow$ | $H \uparrow$ | $Acc_E \downarrow$ | $Acc_L \uparrow$ | $H \uparrow$ | $Acc_E \downarrow$ | $Acc_L \uparrow$ | $H \uparrow$ |
| ESD (Gandikota et al., 2023) | ICCV23 | 52.80 | 66.20 | 55.11 | 18.80 | 68.60 | 74.37 | 51.60 | 68.80 | 56.82 | 49.20 | 66.20 | 57.48 |
| UCE (Gandikota et al., 2024) | WACV24 | 55.70 | 67.40 | 53.46 | 33.60 | 67.80 | 67.09 | 68.00 | 74.69 | 44.80 | 66.80 | 67.60 | 44.53 |
| AP (Bui et al., 2024) | NeurIPS24 | 55.60 | 67.00 | 53.40 | 32.80 | 66.40 | 66.79 | 62.80 | 65.40 | 47.42 | 67.20 | 69.20 | 44.50 |
| AGE (Bui et al., 2025) | ICLR25 | 49.60 | 68.60 | 55.92 | 26.00 | 68.60 | 71.19 | 65.20 | 66.20 | 45.61 | 74.00 | 67.00 | 44.50 |
| AdaVD (Wang et al., 2024) | CVPR25 | 78.40 | 44.65 | 29.11 | 57.20 | 45.65 | 44.17 | 65.60 | 44.30 | 38.72 | 78.80 | 75.83 | 33.13 |
| ACE (Wang et al., 2025) | CVPR25 | 76.00 | 70.22 | 35.77 | 52.80 | 64.60 | 54.54 | 73.20 | 67.00 | 38.28 | 77.20 | 65.60 | 33.83 |
| **Ours** | – | 10.00 | 67.20 | **76.94** | 2.40 | 64.60 | **77.74** | 8.40 | 66.40 | **76.99** | 14.80 | 66.40 | **74.63** |

term $\mathcal{L}_2$, minimized with respect to $\theta'$, ensures that the model preserves its capability on the remaining categories. Furthermore, minimizing $\mathcal{L}_3$ with respect to $\theta'$ guarantees the erasure of the related sub categories. The objective $\mathcal{L}_{\text{lip}}$ minimizes the difference in model outputs between the target category and its perturbed variants, thereby encouraging smoother model behavior. This regularization promotes the forgetting of target concepts while preserving the model's generalization.

## 5 EXPERIMENTS

This section begins with quantitative experiments to evaluate SURE against several baseline methods, followed by qualitative comparisons and visualizations to further validate our approach. We also conduct ablation studies to analyze the effectiveness of our method in detail (Appendix B), including Component Analysis, SURE's performance on multi-subcategory erasure, the effect of perturbation strength, the contribution of Lipschitz regularization, results with Stable Diffusion version 2 and additional analyses in appendix B. Implementation details are provided in Appendix A.

### 5.1 OBJECT RELATED CONCEPTS

**Datasets:** To evaluate the effectiveness of SURE on object-type categories, we curate four distinct categories, each containing five target subcategories. We focus on three key properties for object-related category erasure. (a) *Singularity*: We select categories with distinct and well-defined subcategories—*Guns*, *Blade Weapons*, *Musical Instruments*, and *Toys*—as the target category for unlearning. (b) *Efficacy*: We assess whether subcategories can be effectively erased using simple prompts in the format "A photo of {sub category}". The specific subcategories for each category are: Guns: *SMG, Rifle, Pistol, Shotgun, Revolver*; Blade Weapons: *Knife, Sword, Spear, Shotel, Dagger*; Musical Instruments: *Trumpet, Guitar, Drums, Saxophone, Piano*; Toys: *Stuffed Toy, Doll, Lego Toy, Toy Car, Funko Toy*. We generate 50 prompts for each subcategories, resulting in a total of 250 prompts per category. (c) *Locality*: To ensure that erasing target concepts does not affect unrelated ones, we evaluate SURE on ten non-target classes from CIFAR-10 (Krizhevsky et al., 2009). For each class, we generate 50 paraphrased prompts to simulate real-world use cases where prompts are often descriptive and may not explicitly mention the concept. For example, a paraphrased prompt for "dog" could be: "a Doberman Pinscher in a police vest, part of a K-9 unit." Our dataset curation uses WordNet to construct our object category erasure dataset. Further details regarding the dataset and its selection criteria are provided in Appendix A.2.

**Evaluation Metrics:** To assess the effectiveness of category erasure, we follow prior work (Huang et al., 2024; Lu et al., 2024) and utilize GroundingDINO (Liu et al., 2024) to determine whether the specified target category is present in the generated images. This enables us to measure the accuracy on erased category ($Acc_E$) and on non-erased (remaining) category ($Acc_L$). To calculate $Acc_E$, we use subcategories for each category. For example, during training, we designate the category 'Guns' as the target for erasure. In the evaluation phase, we generate images using subcategories such as Gun, SMG, Rifle, Pistol, Shotgun, and Revolver. We then apply GroundingDINO to detect the presence of these subcategories. The results are reported as the average percentage of images in which subcategories appear when we input prompts. A lower $Acc_E$ indicates better erasure efficacy, while a higher $Acc_L$ signifies stronger Locality. To provide a unified evaluation, we adopt the harmonic mean $H$ between ($100 - Acc_E$) and $Acc_L$, following the metric proposed in (Lu et al., 2024; Lee et al., 2025).

**Experiment Results:** In Table 1, we present the results of SURE across four categories: Guns, Blade Weapons, Musical Instruments, and Toys. Notably, SURE achieves the highest harmonic

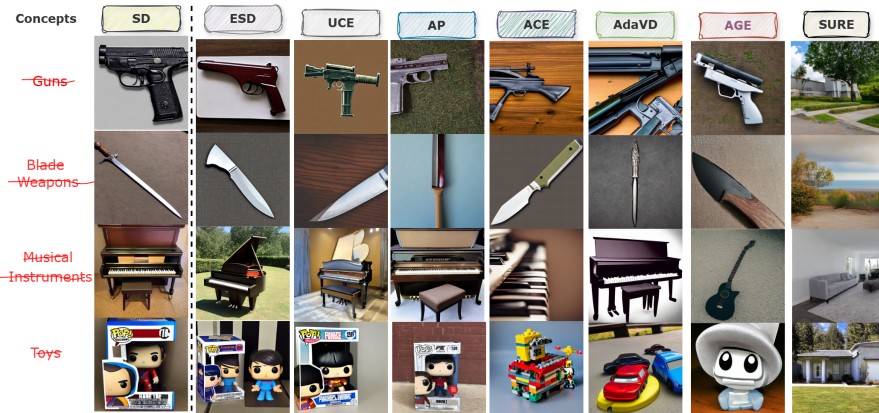

Figure 4: Visual comparison of erasing effectiveness on object-related concepts. It is evident that the compared methods do not completely eliminate the entire target category. In contrast, SURE maps the erased objects to neutral concepts, ensuring effective removal of object-related concepts associated with the category.

mean across all four categories, demonstrating its effectiveness in erasing an entire category (including subcategories) using only the parent category label. This indicates a strong balance between effective removal of category and maintaining sufficient unrelated content in the locality. Specifically, SURE outperforms the second-best method by a relative percentage difference of 31.64%, 8.79%, 30.14%, and 25.96% across the guns, blade weapons, musical instruments, and toys categories, respectively. Some baseline methods (Wang et al., 2024; 2025) demonstrate effectiveness in erasing individual instances of a concept, as reported in their respective studies. However, when tasked with removing an entire category, these methods do not consistently eliminate all related concepts within that category, as shown in Table 1. In contrast, SURE demonstrates robust categorical forgetting, as reflected by the low $\text{Acc}_E$ and comparable $\text{Acc}_L$, achieving an optimal balance between erasure efficacy and locality. We additionally visualize examples of erasing different categories in Figure 4. As observed in the figure, the compared methods fail to erase the subcategories belonging to the parent category, with their outputs closely resembling the original images generated by the pre-trained Stable Diffusion model. In contrast, SURE successfully erases all subcategories associated with the same parent category.

## 5.2 EXPLICIT CONTENTS ERASURE

**Datasets:** To evaluate the proposed method on explicit concept erasure and the unlearning of Not-Safe-For-Work (NSFW) content, we utilize the Image-to-Prompt (I2P) dataset (Schramowski et al., 2023), which contains 4,703 prompts covering sexual, violent, and racist content. Unlike object concept erasure—where categories are clearly defined—we follow prior works and select "Nudity" as the category to be removed from the text-to-image (T2I) model. As suggested by Gandikota et al. (Gandikota et al., 2023), effective erasure of such content should be *global* and independent of specific keywords; that is, the model should remove the concept even when it is implied indirectly or referred to using synonyms. We generate images using all 4,703 prompts with the fine-tuned model to assess efficacy. We use the COCO-30K (Lin et al., 2014) validation set to evaluate locality to ensure that unrelated, safe content remains unaffected.

**Evaluation Metrics:** To evaluate the effectiveness of concept erasure and the presence of nudity in generated images, we employ NudeNet (Bedapudi, 2019), which detects various types of exposed body regions. The model outputs multi-label predictions with associated confidence scores. We use the Nudity Exposure Rate (NER-k) metric (Bui et al., 2024), defined as the proportion of images in which any exposed body parts are detected with a confidence score above a given threshold $k$. To assess content locality and preservation, we report the Frechet Inception Distance (FID) on the COCO-30K validation set.

**Experiment Results:** We report the explicit content erasure performance in Table 2. It is evident from the results that our method outperforms all baselines, achieving Nudity Exposure Rates of 1.70, 0.47, 0.11, and 0.01 at thresholds of 0.3, 0.5, 0.7, and 0.8, respectively. This corresponds to relative percentage difference of 72.65%, 106%, 97.67%, and 120% compared to the second-best

Table 2: Comparison of text-to-image erasure models on the I2P dataset in the nudity erasure setting, evaluated using NER and FID metrics (lower is better ↓).

| Model | Venue | NER-0.3 ↓ | NER-0.5 ↓ | NER-0.7 ↓ | NER-0.8 ↓ | FID ↓ |
|---|---|---|---|---|---|---|
| CA (Kumari et al., 2023) | CVPR 2023 | 13.84 | 9.27 | 4.74 | 1.68 | 20.76 |
| ESD (Gandikota et al., 2023) | ICCV 2023 | 5.32 | 2.36 | 0.74 | 0.23 | 17.14 |
| UCE (Gandikota et al., 2024) | WACV 2024 | 6.87 | 3.42 | 0.68 | 0.21 | 15.98 |
| AP (Bui et al., 2024) | NeurIPS 2024 | 3.64 | 1.70 | 0.40 | 0.06 | 15.52 |
| AGE (Bui et al., 2025) | ICLR 2025 | 5.06 | 1.53 | 0.32 | 0.04 | 14.20 |
| **SURE (ours)** | – | **1.70** | **0.47** | **0.11** | **0.01** | **14.02** |

Table 3: Comparison of CLIP and LPIPS scores across different methods (lower CLIP and higher LPIPS indicate better erasure), evaluated on five different artists style.

| Model | Venue | CLIP Score ↓ | LPIPS Score ↑ |
|---|---|---|---|
| ESD (Gandikota et al., 2023) | ICCV23 | $23.56 \pm 4.73$ | $0.72 \pm 0.11$ |
| CA (Kumari et al., 2023) | CVPR23 | $27.79 \pm 4.67$ | $0.82 \pm 0.07$ |
| UCE (Gandikota et al., 2024) | WACV24 | $24.47 \pm 4.73$ | $0.74 \pm 0.10$ |
| MACE (Lu et al., 2024) | CVPR24 | $27.96 \pm 4.22$ | $0.60 \pm 0.10$ |
| AP (Bui et al., 2024) | NeurIPS24 | $21.57 \pm 5.46$ | $0.78 \pm 0.10$ |
| AGE (Bui et al., 2025) | ICLR25 | $22.44 \pm 5.03$ | $0.80 \pm 0.12$ |
| **SURE (Ours)** | – | $\mathbf{18.02 \pm 5.96}$ | $\mathbf{0.85 \pm 0.10}$ |

method at each threshold. These results demonstrate the effectiveness of our approach in preventing the generation of inappropriate content. Additionally, our method achieves an FID score of 14.02 on the COCO-30K validation set, indicating successful preservation of non-target concepts. Additional results are provided in the appendix B.

## 5.3 ARTISTIC STYLE ERASURE

**Datasets:** To assess the performance of our approach and baseline methods in removing multiple artistic styles from Stable Diffusion v1.4, we select five prominent artists: *Kelly McKernan*, *Thomas Kinkade*, *Tyler Edlin*, *Kilian Eng*, and *Ajin Demi Human*. For fine-tuning, only the artist's names are used as the target concepts. During evaluation, we employ a curated set of detailed prompts (Bui et al., 2024; 2025) tailored to each artist, rather than generic descriptions (Gandikota et al., 2023). Each prompt is combined with five random seeds, resulting in the generation of 200 images per artist for every method.

**Evaluation Metrics:** To evaluate the removal of artistic styles, we use CLIP (Radford et al., 2021) to calculate the similarity between each generated image and its associated text prompt, where lower similarity indicates more effective erasure of the target style. Additionally, we utilize LPIPS (Zhang et al., 2018) to quantify perceptual differences between images produced by the original Stable Diffusion model and those generated by edited models. A lower LPIPS score signifies less distortion. For erased concepts, a higher LPIPS score indicates more effective removal.

**Experiment Results:** The evaluation results are presented in Table 3. It is evident that our method outperforms existing approaches in both CLIP and LPIPS scores. Specifically, our method achieves an average CLIP score of 18.02, reflecting a 17.93% relative improvement over the second-best method, AP. Additionally, SURE obtains the highest LPIPS score of 0.85, indicating superior perceptual quality and more effective erasure of artistic styles compared to other baselines.

## 6 CONCLUSION

Our proposed novel approach, SURE, effectively addresses the task of removing entire sub categories from pre-trained text-to-image diffusion models using only a single parent category. It leverages Lipschitz regularization to promote smooth model behavior around target category and incorporates a neutral eraser and preservation objective to balance effective concept forgetting with the retention of unrelated content. SURE consistently outperforms existing methods in balancing erasure and preservation through comprehensive experiments across object category removal, explicit content suppression, and artistic style elimination. In future work, we aim to further enhance non-target concept preservation while maintaining the ability to remove entire categories effectively.

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

# Appendix

## Table of Contents

## A  FURTHER IMPLEMENTATION DETAILS

In this section, we provide more details on the proposed Category Unlearning via Regularized Erasure (SURE). In the following subsections, we present details on Lipschitz regularization, the search for related concepts, and the computational analysis.

### A.1  IMPLEMENTATION DETAILS

For a fair comparison, we use publicly available open-source implementations of all baseline methods, adopting their recommended hyperparameter settings. Since several baselines are specifically designed for Stable Diffusion (SD-1.4), all experiments are conducted using this version to maintain consistency across methods. Image generation is performed with the DDIM sampler using 50 sampling steps and a guidance scale of 7.5, producing outputs at a resolution of $512 \times 512$ pixels. All experiments are run on a single NVIDIA A6000 GPU with 48 GB of VRAM. We compare our method, SURE, with six state-of-the-art concept-erasing approaches: Erased Stable Diffusion (ESD) (Gandikota et al., 2023), Unified Concept Erasure (UCE) (Gandikota et al., 2024), Adversarial Preservation (AP) (Bui et al., 2024), Adaptive Guided Erasure (AGE) (Bui et al., 2025), Adaptive Value Decomposer (Ada VD) (Wang et al., 2024), and Anti-Editing Concept Erasure (ACE). For our approach, we follow the experimental configurations adopted in prior work (Wu et al., 2024; Bui et al., 2024; 2025). Specifically, we fine-tune Stable Diffusion 1.4 for 1000 steps using a batch size of 1, the Adam optimizer, and a learning rate of $10^{-5}$.

## A.2 DATASETS AND WORDNET ONTOLOGY

We use the WordNet [1] to construct our object category erasure dataset. WordNet provides a hierarchical lexical structure organized by *hypernyms* and *hyponyms*, where hypernyms denote category and hyponyms represent subcategory (i.e., "is-a" relationships). By querying the WordNet 3.0, we can retrieve direct subcategories, which enables us to systematically build category–subcategory.

For evaluation, we selected **Guns** and **Bladed Weapons** as sensitive and harmful categories. Additionally, we included **Musical Instruments** and **Toys** as balanced, non-sensitive categories to test whether our method generalizes beyond harmful concepts. We argue that this curated benchmark provides a more rigorous and realistic testbed for category-level erasure.

For example, Imagenette includes only ten broad categories (e.g., "church"), but it lacks sufficient subcategory variation necessary for evaluating category-wide erasure. In contrast, WordNet offers hierarchical structure, which is essential for our setting.

The subcategories for each chosen category were selected directly from the WordNet ontology by retrieving top-level hyponyms. For instance, within the **Guns** category (hypernym), WordNet provides the following sub categories:

- Rifle
- Submachine Gun
- Shotgun
- Pistol
- Revolver

In our dataset, we retain the **subcategory level** (e.g., *Rifle*, *SMG*, *Pistol*, *Shotgun*, *Revolver*), and exclude specific object instances (e.g., "Colt" or "Luger"). A similar data curation strategy is applied for **Bladed Weapons**, **Toys**, and **Musical Instruments**.

## A.3 LIPSCHITZ REGULARIZATION

We adopt Lipschitz regularization for concept erasure in diffusion models to enable the forgetting of concepts from specific categories. Our regularization leverages the principle of Lipschitz continuity as a mechanism for unlearning. The core idea involves adding Gaussian noise to the input concept image targeted for erasure and minimizing the ratio of output differences between the perturbed and original inputs, relative to the input perturbation. This form of regularization was originally introduced by Yoshida et al. Yoshida & Miyato (2017) to enhance generalization. However, Foster et al. (2024) showed that, with sufficiently strong Gaussian perturbations, the model's learned response to the targeted input can be effectively removed.

Specifically, for each concept to be erased, perturbations are applied to the generated concept image to produce new, semantically similar variants. The model is then optimized using a weighted objective (ref. eq. 10) that minimizes the output discrepancy between the original concept image marked for erasure and its perturbed counterparts (ref Fig. 5). This targeted smoothing encourages the model to unlearn the specified concept while preserving its overall generalization performance.

Let $x$ be an image generated by the model conditioned on a concept embedding $c_e$. We apply a small perturbation $\xi$ to $x$, where $\xi$ is independently sampled from a Gaussian distribution, i.e., $\xi \sim \mathcal{N}(0, \sigma^2)$. This results in a perturbed image $x' = x + \xi$. Both $x$ and $x'$ are passed through the first-stage VAE encoder $f_{\theta'}(\cdot)$ to obtain their latent embeddings. The Lipschitz regularization loss is defined as follows:

$$\mathcal{L}_{lip} = \mathbb{E}\left(\frac{\|f_{\theta'}(x) - f_{\theta'}(x + \xi)\|_2}{\|x - (x + \xi)\|_2}\right) \approx \frac{1}{N}\sum_{i=1}^{N}\left(\frac{\|f_{\theta'}(x) - f_{\theta'}(x + \xi_i)\|_2}{\|\xi_i\|_2}\right), \quad (10)$$

where $\xi_i$ is the $i$-th perturbation sample and $N$ is the total number of perturbation samples. Lipschitz regularization is applied to the VAE encoder's latent output, where the input is a perturbed version

---

[1]https://wordnet.princeton.edu/

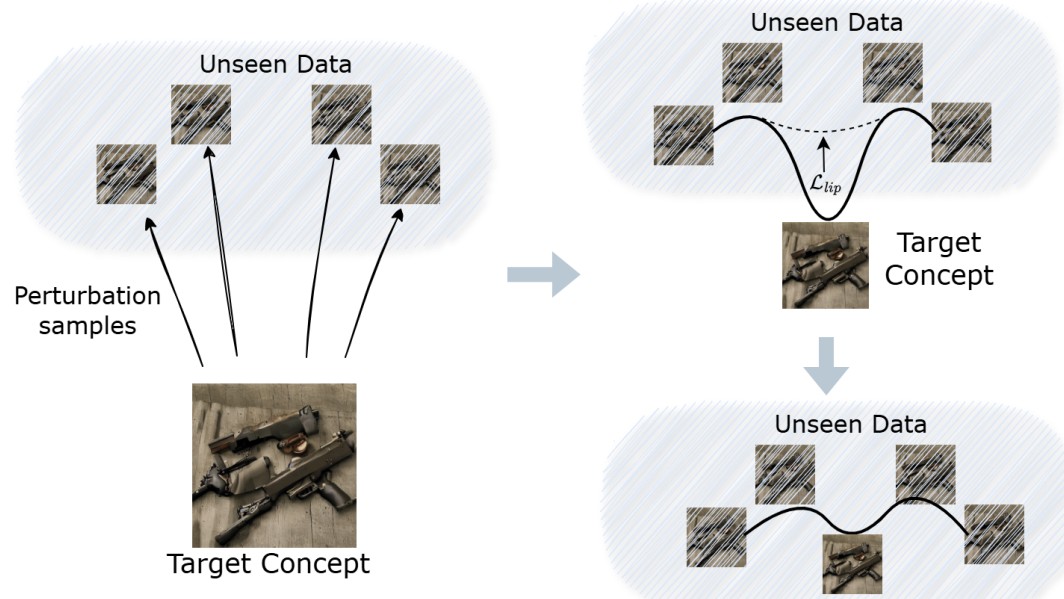

Figure 5: Illustration of the target concept erasure process. An image is generated by the model conditioned on a concept embedding. A perturbation, independently sampled from a Gaussian distribution, is then applied to the image, resulting in perturbed image samples (N). The model is optimized using a weighted objective that minimizes the output discrepancy between the original concept image marked for erasure and its perturbed counterparts. This targeted smoothing encourages the model to unlearn the specified concept while preserving its overall generalization performance.

of the image generated by the model, conditioned on a concept embedding (i.e., the output of the VAE decoder). During training, the VAE encoder is reused to compute this regularization output. The encoder maps the noisy perturbed images into the latent space, where we measure the sensitivity to these perturbations. The model is then updated using the weighted objective described in Eq. 8 of the main paper. This coupling in the intermediate latent space guides the output of the diffusion process toward concept erasure behavior (e.g., in Table 4). This regularization leads to performance gains and more effective category-level forgetting.

SURE operates in the latent diffusion framework, where the diffusion U-Net is trained to denoise latents generated by the (frozen) VAE encoder. Smoothness in the encoded space therefore directly corresponds to smoothness in the U-Net's feature space, because every forward/backward pass begins with the encoded representation and the U-Net processes only these latents. Enforcing Lipschitz continuity in this space ensures that small perturbations in the latent input region corresponding to the erased category yield small changes in the U-Net output, which suppresses subcategorical memorization. As shown in the Lipschitz loss in Eq. (8) (main paper) and Eq. 10 (Appendix), the regularization term where $f_{\theta'}$ denotes the frozen VAE encoder followed by the *trainable* diffusion UNet denoiser. Because the encoder is fixed, no gradients propagate through it; all gradients from $\mathcal{L}_{\text{lip}}$ flow exclusively into the UNet parameters. A detailed algorithm of the SURE training is provided in Algorithm 1.

### A.4 SEARCHING FOR RELATED CONCEPTS

To identify related concepts, we utilize the CLIP vocabulary, which consists of 49,408 tokens. Although this vocabulary is extensive, it includes a considerable number of nonsensical or meaningless tokens (e.g., "...", ".¡/w¿"). Therefore, we filter out such tokens to ensure the quality and relevance of the search space. For computational efficiency, we further limit the related concept candidates to the top 100 most similar words to the target concept. As a result, the concept space used for searching related concepts is restricted to just 100 words. Another motivation for using CLIP is to ensure that the concept space captures semantically related concepts rather than subcategories. For example,

---

**Algorithm 1** Subcategorical Unlearning via Regularized Erasure

---

**Input:** Pretrained model $\epsilon_\theta$, parent category $c_e$, neutral concept $c_n$, concept space $C$, loss weights $\lambda$
**Output:** Eraser model $\epsilon_{\theta'}$
Initialize eraser model parameters: $\theta' \leftarrow \theta$   Precompute CLIP embeddings for concept space $C$ and
  construct candidate pool $C \setminus \mathbf{E}$
**for** *each training step* **do**
  Sample $c_e$ and $c_n$
  // (1) Erasure loss $\mathcal{L}_1 \leftarrow \|\epsilon_{\theta'}(\tau(c_e)) - \epsilon_\theta(\tau(c_n))\|_2^2$
  // (2) Neutral preservation loss $\mathcal{L}_2 \leftarrow \|\epsilon_{\theta'}(\tau(c_n)) - \epsilon_\theta(\tau(c_n))\|_2^2$
  // (3) Related subcategory discovery and erasure Select $c_r$ from $C \setminus \mathbf{E}$ via Gumbel–Softmax
    maximizing $\mathcal{L}_3 = \|\epsilon_{\theta'}(\tau(c_r)) - \epsilon_\theta(\tau(c_n))\|_2^2$
  // (4) Lipschitz reg. Generate $x$; sample $\xi \sim \mathcal{N}(0, \sigma^2 I)$, form $x' = x + \xi$   Compute $\mathcal{L}_{\text{lip}}$
  // Final update $\mathcal{L}_{\text{total}} \leftarrow \mathcal{L}_1 + \mathcal{L}_2 + \mathcal{L}_3 + \lambda\mathcal{L}_{\text{lip}}$   Update $\theta'$ by minimizing $\mathcal{L}_{\text{total}}$
**end**
**return** $\epsilon_{\theta'}$

---

the semantic category of *gun* may include related concepts such as *violence*, *war*, and others, which are contextually associated within the concept space.

Since the entire concept space is discrete in nature, exhaustively searching for concepts across the whole space is computationally challenging. To enable differentiable selection over a discrete set of related concepts during training, we experimented with the Straight-Through Estimator (STE), Sparsemax / Entmax, and the Gumbel-Softmax operator Bui et al. (2024). Based on the results in Table 6, we selected Gumbel-Softmax to approximate sampling from a categorical distribution in a fully differentiable manner.

We begin by precomputing the text embeddings $E_\mathcal{C}$ of the concept space $C$ using the text encoder $\tau(\cdot)$ from the foundation model. The representation of a related concept, $\tau(c_r)$, is then computed as the inner product of $D(\beta)$ and $E_\mathcal{C}$, where $\beta \in \mathbb{R}^{|C|}$ is a learnable variable.

$$\max_\beta \underbrace{\|\epsilon_{\theta'}(\mathbf{D}(\beta) \odot E_\mathcal{C}) - \epsilon_\theta(\tau(c_n))\|_2^2}_{\mathcal{L}_3} \tag{11}$$

Here D is the differenciable operator use to make the training process differeneciable. we seleect the Gumbel-Softmax operator with a temperature below 1 to ensure that only a few concepts are selected as the related concept. we optimize the following objective by maximizing $\mathcal{L}_3$ (ref. Eq. 11) with respect to $c_r$. In other words, it seeks a concept $c_r$ whose output deviates the most from the neutral concept $c_n$.

## A.5   COMPUTATIONAL ANALYSIS

The average training time of SURE per iteration is 6.42 seconds, with a GPU memory usage of approximately 31,010 MB. The inference time of the trained model is around 9 seconds for image generation, requiring approximately 7,716 MB of GPU memory. Furthermore, by using only a small subset of words from the CLIP vocabulary, we reduce the overhead associated with the concept space, making our approach more computationally efficient. The average training time per iteration with Lipschitz regularization is 6.42 seconds, compared to 6.26 seconds without it, a slight increase of only 2.5%. GPU memory usage remains nearly identical: 31,010 MB with Lipschitz vs. 30,997 MB without. The model performs identically at inference time to standard fine-tuned diffusion models, requiring 9 seconds per image and 7,716 MB of GPU memory.

## A.6   THEORETICAL JUSTIFICATION OF LIPSCHITZ REGULARIZATION

In this subsection, we demonstrate how Lipschitz smoothness reduces memorization. Our justification adapts the unlearning framework of Xu et. al. (Xu & Strohmer, 2025) to the concept level in diffusion models.

**Definition 1** ($\varepsilon$-Differential Concept Unlearning). *Let $c_e$ denote the erased concept (e.g., "gun"), and let $c_n$ denote a neutral concept. Let $f_\theta(c)$ denote the output distribution of a text-to-image diffusion model conditioned on the embedding of concept $c$. A fine-tuned eraser model $f'\theta$ satisfies $\varepsilon$-Differential Concept Unlearning ($\varepsilon$-DCU) if, for all outputs $y$,*

$$\left| \log \frac{f'\theta(y \mid c_e)}{f'_\theta(y \mid c_n)} \right| \leq \varepsilon. \tag{12}$$

*That is, observing the model output provides negligible evidence about whether the model was conditioned on the erased concept $c_e$.*

**Necessary Condition for Exact Unlearning.** Under mild regularity assumptions, one can show that any model that attempts to reproduce the exact unlearning outcome via retraining can violate $\varepsilon$-DCU only if it becomes *non-smooth*, i.e., its Lipschitz constant must exceed a threshold. This implies that failure of $\varepsilon$-DCU necessarily corresponds to a highly non-Lipschitz, poorly generalizable model (see Lemma 1). Thus, $\varepsilon$-DCU serves as a fundamental requirement for exact unlearning: a model that remains smooth (Lipschitz) cannot simultaneously continue to memorize the erased concept.

**Lemma 1** (Concept Unlearning Requires Lipschitz Smoothness). *Let $f'_\theta$ be a fine-tuned erasure model such that the mapping $c \mapsto f'_\theta(c)$ is L-Lipschitz under a metric $d_Z$ on the concept-embedding space,*

$$W(f'_\theta(c), f'_\theta(c')) \leq L \, d_Z(c, c'), \tag{13}$$

*where $W(\cdot, \cdot)$ denotes the Wasserstein distance between output distributions. Assume that $f'_\theta(c_e)$ and $f'_\theta(c_n)$ admit continuous densities. If $\varepsilon$-DCU is violated, i.e., there exists $y^\star$ such that*

$$\left| \log \frac{f'_\theta(y^\star \mid c_e)}{f'_\theta(y^\star \mid c_n)} \right| > \varepsilon, \tag{14}$$

*then the Lipschitz constant must satisfy*

$$L > L^*(\varepsilon) := \frac{\delta \, (e^\varepsilon - 1) \min\{f'_\theta(y^\star \mid c_e), \, f'_\theta(y^\star \mid c_n)\}}{d_Z(c_e, c_n)}, \tag{15}$$

*for some $\delta > 0$ such that this density gap persists on the entire ball $B_\delta(y^\star)$.*

*Proof.* Since $\varepsilon$-DCU is violated, continuity of the densities implies the existence of $\delta > 0$ such that for all $y \in B_\delta(y^\star)$,

$$f'_\theta(y \mid c_e) \geq e^\varepsilon \, f'_\theta(y \mid c_n). \tag{16}$$

Hence the pointwise density difference is bounded below by

$$f'_\theta(y \mid c_e) - f'_\theta(y \mid c_n) \geq (e^\varepsilon - 1) \min\{f'_\theta(y \mid c_e), \, f'_\theta(y \mid c_n)\}. \tag{17}$$

Since $f'_\theta(c_e)$ and $f'_\theta(c_n)$ admit densities, the Wasserstein distance satisfies

$$W(f'_\theta(c_e), f'_\theta(c_n)) \geq \delta \int_{B_\delta(y^\star)} \left| f'_\theta(y \mid c_e) - f'_\theta(y \mid c_n) \right| dy. \tag{18}$$

Using the lower bound on the density difference on $B_\delta(y^\star)$ gives

$$W(f'_\theta(c_e), f'_\theta(c_n)) \geq \delta \, (e^\varepsilon - 1) \min\{f'_\theta(y^\star \mid c_e), \, f'_\theta(y^\star \mid c_n)\}. \tag{19}$$

On the other hand, Lipschitz continuity implies

$$W(f'_\theta(c_e), f'_\theta(c_n)) \leq L \, d_Z(c_e, c_n). \tag{20}$$

Combining the upper and lower bounds yields

$$L \, d_Z(c_e, c_n) \geq \delta \, (e^\varepsilon - 1) \min\{f'_\theta(y^\star \mid c_e), \, f'_\theta(y^\star \mid c_n)\}, \tag{21}$$

$\square$

which can also be rearranges to $L > L^*(\varepsilon)$ .

Lemma 1 shows that a model which still exhibits distinguishable behavior for the erased concept must necessarily operate with a large Lipschitz constant. Thus, enforcing small local Lipschitz constants in the concept latent space. The Lipschitz regularization $\mathcal{L}_{lip}$ term in SURE promotes this smoothens and thereby achieving effective concept forgetting.

Table 4: Ablation studies on the components of SURE (left) and sensitivity to the number of Gaussian perturbation samples ($n$) for the **Guns** category (right).

| Method | $Acc_E$ ($\downarrow$) | $Acc_L$ ($\uparrow$) |
|---|---|---|
| **SURE** | **10.00** | **67.20** |
| w/o Lipschitz | 25.20 | 65.80 |
| w/o Preservation | 30.00 | 64.80 |

| Samples $n$ | $Acc_E$ ($\downarrow$) | $Acc_L$ ($\uparrow$) |
|---|---|---|
| $n = 1$ | 19.60 | 65.40 |
| $n = 3$ | 30.40 | 67.00 |
| $n = 5$ | **10.00** | **67.20** |
| $n = 7$ | 27.60 | 66.60 |

## B  ADDITIONAL EXPERIMENTS

We conducted ablation studies to analyze the effectiveness and sensitivity of SURE. Additional ablation experiments, including results with Stable Diffusion version 2, are also priovided.

### B.1  COMPONENT ANALYSIS:

We assess the individual contribution of Lipschitz and the Neutral preservation loss (ref. Table 4 left). When the Lipschitz regularization is removed (w/o Lip), the model's ability to erase the target concept significantly degrades, as shown by a higher $Acc_E$ (25.20). Removing the preservation objective (w/o Preservation) further reduces erasure performance ($Acc_E = 30.00$) and slightly affects the model's retention of non-target concepts ($Acc_L = 64.80$). Additionally, we have included an ablation in Table 5 where the Lipschitz regularization term is removed to evaluate its impact across multiple semantic categories. The results clearly show that Lipschitz regularization is essential for preventing subcategorical memorization: removing it leads to a substantial increase in $Acc_E$ across all categories, confirming its effectiveness in achieving category-level erasure.

### B.2  EFFECT OF NUMBER OF PERTURBATION SAMPLES ($n$):

We analyze the sensitivity of SURE to the number of Gaussian noise samples used in the Lipschitz regularization. As shown in Table 4 (right), performance varies with different values of $n$. Increasing $n$ improves the erasure quality, peaking at $n = 5$ with $Acc_E = 10.00\%$ and $Acc_L = 67.20\%$.

The Lipschitz is a Monte Carlo estimate of an expectation over perturbations. When $n$ is very small (e.g., $n = 1$), this estimate has high variance, resulting in a noisy smoothing signal; as a consequence, the model can still retain information about the erased category, leading to higher $Acc_E$. Increasing the number of samples reduces this variance and provides a more stable smoothing effect, consistent with observations in Foster et al. (2024). Setting $n = 5$ offers the most stable behavior around the target category, which empirically yields the strongest forgetting.

For larger $n$, the effective weight of the Lipschitz term becomes too dominate relative to the erasure and preservation losses. Because the loss contribution is kept fixed, adding more perturbation samples increases the overall strength of the smoothing constraint, placing pressure on the model to behave nearly identically across many perturbation directions around the target category.

This "over-smoothing" restricts the optimization too strongly: the model is encouraged to maintain local invariance rather than shifting sufficiently toward the neutral concept required for full erasure ($\mathcal{L}_1$). Empirically, this leads to a decline in forgetting performance (higher $Acc_E$), even though locality ($Acc_L$) remains largely unchanged. Adding too many perturbation samples makes the Lipschitz regularizer so dominant that it starts to show less category-level erasure then the optimal perturbation samples $n$.

### B.3  DIFFERENTIABLE OPERATORS

To enable differentiable training and transform the discrete concept space into a continuous one, we utilize differentiable operators for learning the embedding distribution $\beta$. We conduct an ablation study using four different operators: Gumbel-Softmax, Straight-Through Estimator, Entmax, and Sparsemax. The results are presented in Table 6. Among these, Gumbel-Softmax provides the most efficient training performance by enabling a fully differentiable and continuous selection process for related concepts.

Table 5: Effect of removing Lipschitz regularization across three semantic categories.

| Category | Method | AccE ($\downarrow$) | AccL ($\uparrow$) |
|---|---|---|---|
| Blade Weapons | CURE | 2.40 | 64.60 |
| | w/o Lipschitz | 26.00 | 61.60 |
| Musical Instruments | CURE | 8.40 | 66.40 |
| | w/o Lipschitz | 65.20 | 60.20 |
| Toys | CURE | 14.80 | 66.40 |
| | w/o Lipschitz | 74.00 | 59.00 |

Table 6: Ablation study on differentiable selection methods for related concept retrieval (on Guns object level dataset). $Acc_E$ denotes accuracy on the erased concept (lower is better), and $Acc_L$ denotes accuracy on the preserved concept (higher is better).

| Method | $Acc_E$ ($\downarrow$) | $Acc_L$ ($\uparrow$) |
|---|---|---|
| Gumbel-Softmax | 10.00 | 67.20 |
| Straight-Through Estimator | 44.80 | 67.20 |
| Entmax | 38.40 | 67.00 |
| Sparsemax | 46.80 | 64.20 |

Table 7: Ablation study of Lipschitz Regularization strength on forgetting and preservation performance (on Guns object level dataset). $Acc_E$ is the accuracy on the erased concept (lower is better), and $Acc_L$ is the accuracy on the preserved concept (higher is better).

| Lipschitz Regularization | $Acc_E$ ($\downarrow$) | $Acc_L$ ($\uparrow$) |
|---|---|---|
| $\lambda = 0.01$ | **10.00** | **67.20** |
| $\lambda = 0.1$ | 41.60 | 66.40 |
| $\lambda = 1$ | 22.32 | 65.80 |

The goal of $\mathcal{L}_3$ is to identify the most adversarial related concept i.e., the one whose output deviates maximally from the neutral concept. This is analogous to adversarial example selection, where the optimal perturbation is the one maximizing deviation rather than an average over several directions. A hard-max (implemented via low-temperature Gumbel–Softmax) directly approximates this objective and results in better erasure of subcategorical memorization.

Furthermore, Table 5 shows that soft-average operators such as Entmax and Sparsemax, which distribute probability mass across several candidates, substantially weaken forgetting performance (higher $Acc_E$). This indicates that averaging gradients over multiple related concepts make the optimization less effective.

### B.4 CONTRIBUTION OF LIPSCHITZ REGULARIZATION

In the following ablation study, we evaluate the contribution of Lipschitz Regularization. Specifically, we vary the regularization strength and observe its effect on the forgetting accuracy ($Acc_E$) and the preservation accuracy ($Acc_L$). As shown in Table 7, moderate regularization (e.g., $\lambda = 0.01$) yields the best trade-off between effective forgetting and generalization.

### B.5 EFFECT OF PERTURBATION STRENGTH

In this ablation study, we examine the impact of perturbation strength on the erasure of the target semantic category and the preservation of non-target concepts. Specifically, we vary the amount of Gaussian noise added to the input image by adjusting the value of $\sigma$, which represents the standard deviation of the noise distribution. As shown in Table 8, the best performance is achieved when $\sigma = 0.1$, which provides a balanced level of perturbation for effective Lipschitz regularization.

Table 8: Effect of Gaussian noise level ($\sigma$) on erasure ($Acc_E$) and preservation ($Acc_L$) accuracies (on Guns object level dataset). Lower $Acc_E$ and higher $Acc_L$ indicate better performance.

| **Perturbation Strength** ($\sigma$) | $Acc_E$ ($\downarrow$) | $Acc_L$ ($\uparrow$) |
|---|---|---|
| 0.01 | 20.00 | 63.60 |
| 0.10 | 10.00 | 67.20 |
| 0.30 | 11.20 | 65.00 |

Table 9: Comparison of CLIP and LPIPS scores across different methods, evaluated on five different artistic styles. Lower CLIP and higher LPIPS scores indicate better erasure performance. Additionally, higher CLIP scores on the remaining artists indicate better preservation performance.

| Model | Venue | CLIP Score $\downarrow$ | LPIPS Score $\uparrow$ | CLIP Score $\uparrow$ |
|---|---|---|---|---|
| ESD Gandikota et al. (2023) | ICCV23 | $23.56 \pm 4.73$ | $0.72 \pm 0.11$ | $29.63 \pm 3.57$ |
| CA Kumari et al. (2023) | CVPR23 | $27.79 \pm 4.67$ | $0.82 \pm 0.07$ | $29.85 \pm 3.78$ |
| UCE Gandikota et al. (2024) | WACV24 | $24.47 \pm 4.73$ | $0.74 \pm 0.10$ | $30.89 \pm 3.56$ |
| MACE Lu et al. (2024) | CVPR24 | $27.96 \pm 4.22$ | $0.60 \pm 0.10$ | $\mathbf{31.52 \pm 2.91}$ |
| AP Bui et al. (2024) | NeurIPS24 | $21.57 \pm 5.46$ | $0.78 \pm 0.10$ | $30.13 \pm 3.44$ |
| AGE Bui et al. (2025) | ICLR25 | $22.44 \pm 5.03$ | $0.80 \pm 0.12$ | $30.45 \pm 3.35$ |
| **SURE (Ours)** | – | $\mathbf{18.02 \pm 5.96}$ | $\mathbf{0.85 \pm 0.10}$ | $30.91 \pm 3.45$ |

Table 10: Evaluation of SURE's performance in terms of erasure ($Acc_E$) and preservation ($Acc_L$) accuracies using Stable Diffusion 2.

| **Concept** | $Acc_E$ **(SDv2)** | $Acc_L$ **(SDv2)** |
|---|---|---|
| Guns | 18.7 | 68.9 |
| Musical Instruments | 20.4 | 66.8 |
| Blade Weapons | 16.1 | 67.6 |
| Toys | 20.3 | 68.3 |

### B.6 PRESERVATION ON ARTISTS STYLE

In this experiment (ref. Table 9), we assess the effectiveness of our method in erasing artistic style concepts. We select five renowned artists—*Kelly McKernan*, *Thomas Kinkade*, *Tyler Edlin*, *Kilian Eng*, and *Ajin Demi Human*—whose distinctive styles are frequently reproduced by text-to-image generative models. The erasure task involves removing the style associated with one artist while evaluating the preservation of the styles of the remaining four. We use the CLIP score to measure both the success of erasure and the fidelity of preservation. As reported in the Table 9, our method achieves a balanced performance in both aspects. While MACE demonstrates the highest preservation CLIP score, it performs poorly in erasing the target artistic style. In contrast, our method achieves the best erasure performance among all reported baselines, along with a comparable preservation score.

### B.7 RESULTS WITH STABLE DIFFUSION VERSION 2

We experiment with Stable Diffusion v2.1 on object-related concepts such as *Guns*, *Blade Weapons*, *Musical Instruments*, and *Toys*. As shown in the table 10, our method achieves a balanced performance in both erasure and preservation, demonstrating its effectiveness across diverse categories.

### B.8 SURE MULTIPLE CONCEPT ERASURE

In this experiment, we trained the model using only the parent concept (Blade Weapons) and measured the erasure performance across an extended set of blade-related concepts. The results are:

- 5 concepts: AccE = 2.40%

- 10 concepts: AccE = 2.80%

- 15 concepts (Knife, Sword, Spear, Shotel, Dagger, Katana, Machete, Scimitar, Sabre, Rapier, Cutlass, Bayonet, Falchion, Cleaver, Hook Sword): AccE = 4.40%

Table 11: Effect of adding Lipschitz regularization to the ACE baseline on gun dataset. Incorporating the Lipschitz term (ACE+Lip) improves erasure performance (lower AccE) while maintaining comparable locality (AccL), demonstrating that the regularization can be used as a plug-and-play.

| Method | AccE ($\downarrow$) | AccL ($\uparrow$) |
|---|---|---|
| ACE | 76.00 | 70.22 |
| ACE + Lip | 41.00 | 68.70 |

These results show that SURE maintains strong erasure performance even when tested on large set of multiple concepts.

### B.9 LIPSCHITZ REGULARIZATION AS A PLUG-AND-PLAY COMPONENT

Because the regularization operates on the intermediate latent outputs of the diffusion model, it is independent of the specific architectural choices made by existing concept erasure methods. As shown in Table 11, we incorporated the Lipschitz regularization term into the ACE (Wang et al., 2025) baseline. The resulting performance improvements demonstrate that the Lipschitz term can be used as a plug-and-play component in different erasure frameworks.

### B.10 CONCEPT SPACE SIZE

Varying the size of the concept space $C$ for initial pruning results in negligible changes in performance. This robustness arises because the differentiable Gumbel–Softmax operator consistently selects only a small number of high-impact related concepts, regardless of the size of the initial pool. To validate this, we added experiments showing that erasure performance remains stable across different choices of $|C|$. For example, increasing the pool to 200 CLIP tokens from 100 results in only a minor GPU memory overhead of approximately 20 MB, while maintaining similar performance ($\text{Acc}_E = 10.40$, $\text{Acc}_L = 67.80$). Further expanding the pool to 500 CLIP tokens yields comparable results ($\text{Acc}_E = 10.00$, $\text{Acc}_L = 68.00$).

### B.11 BEHAVIOR ON NON-TARGET, SEMANTICALLY ADJACENT CONCEPTS

We also evaluated SURE on concepts that are semantically related to the erased category but are not part of the target hypernym. For example, after erasing "Blade Weapons," we tested the concept "Scissors," which is visually similar yet not a subcategory of the erased class. SURE preserves this concept with $\text{Acc}_E = 94.00\%$, whereas ESD significantly over-erases it ($\text{Acc}_E = 81.00\%$). These results indicate that SURE maintains clear semantic boundaries, even when the erased category spans diverse or visually overlapping regions.

## C QUALITATIVE RESULTS

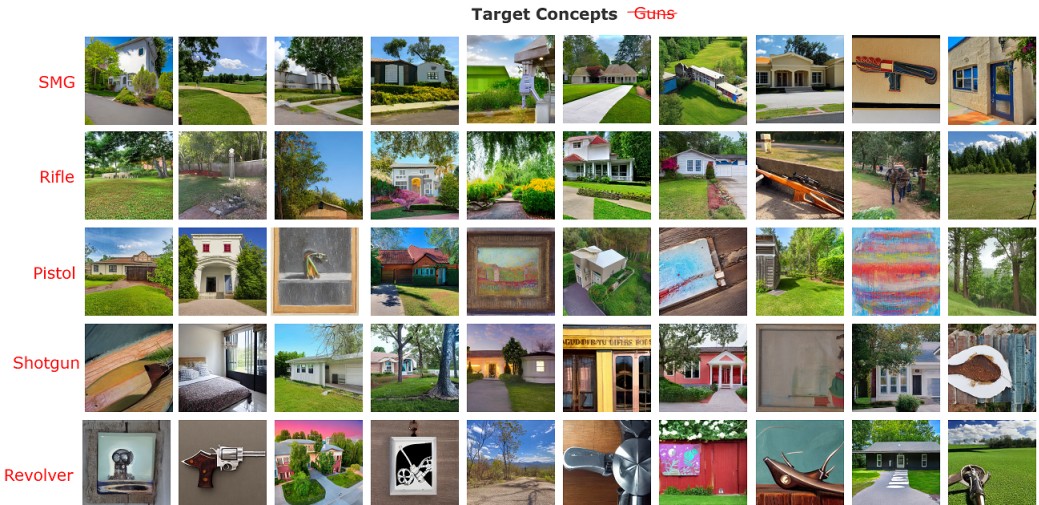

Figure 6: Illustration of the erasure of the *guns* semantic category, along with visualizations of its subcategories (e.g., SMG, rifle, pistol) generated by the erased model. The visual results clearly demonstrate that SURE effectively removes the entire semantic category.

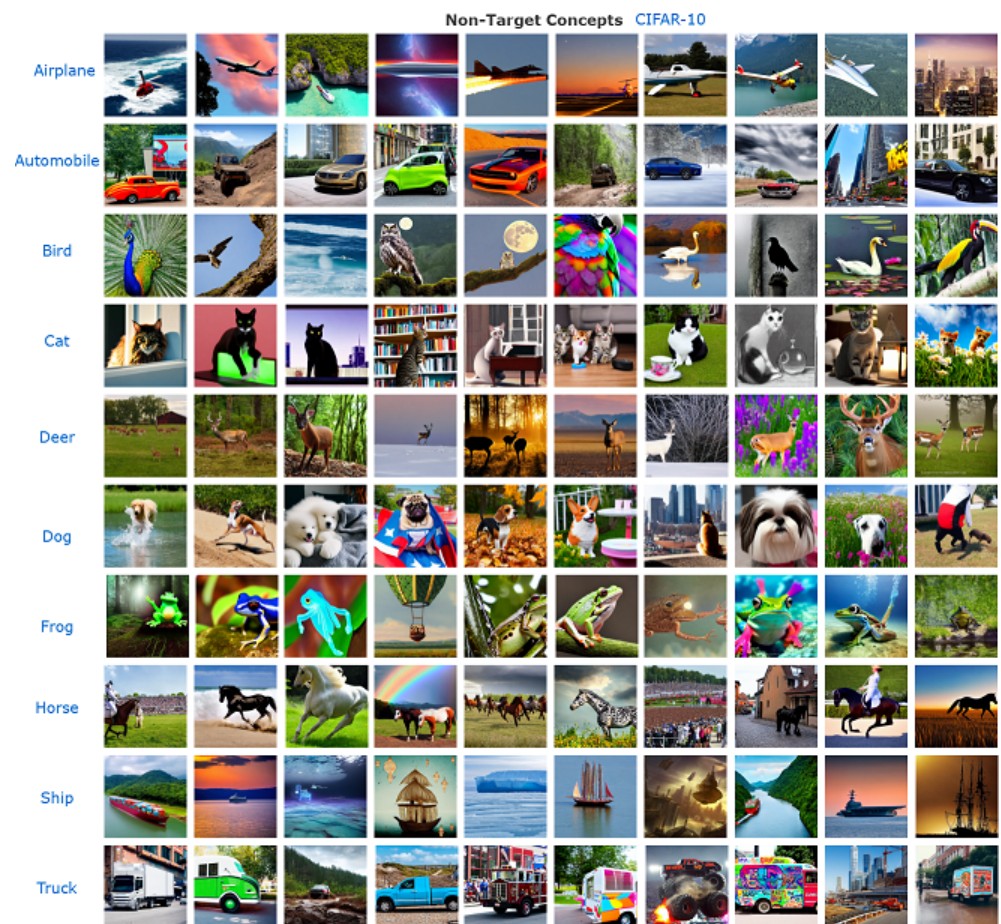

Figure 7: Illustration of the preservation of non-target classes (CIFAR-10) by the model after erasing the *gun* semantic category. The preservation performance is clearly maintained, as all non-target classes are accurately and distinctly generated by the model.

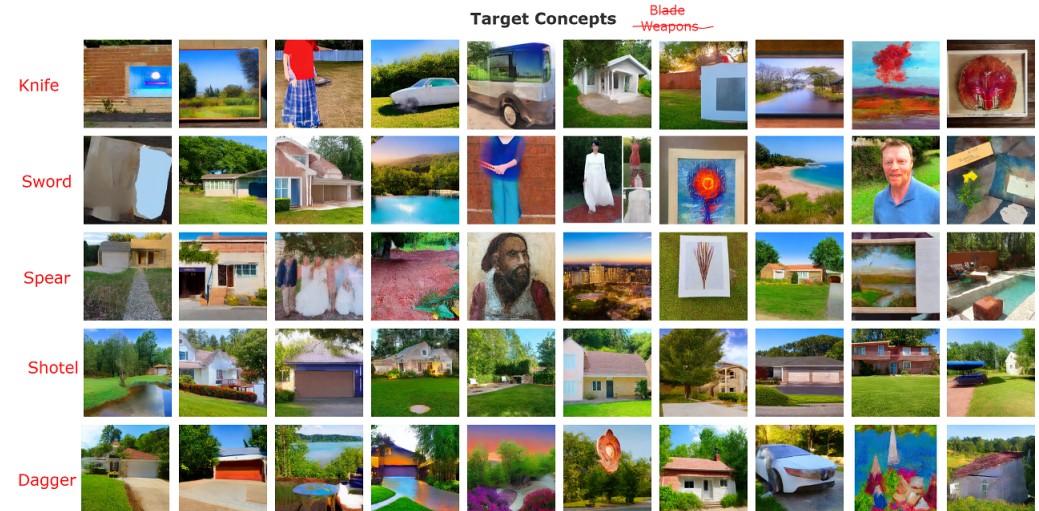

Figure 8: Illustration of the erasure of the *Blade Weapons* semantic category, along with visualizations of its subcategories (e.g., Knife, Sword, Spear) generated by the erased model. The visual results clearly demonstrate that SURE effectively removes the entire semantic category.

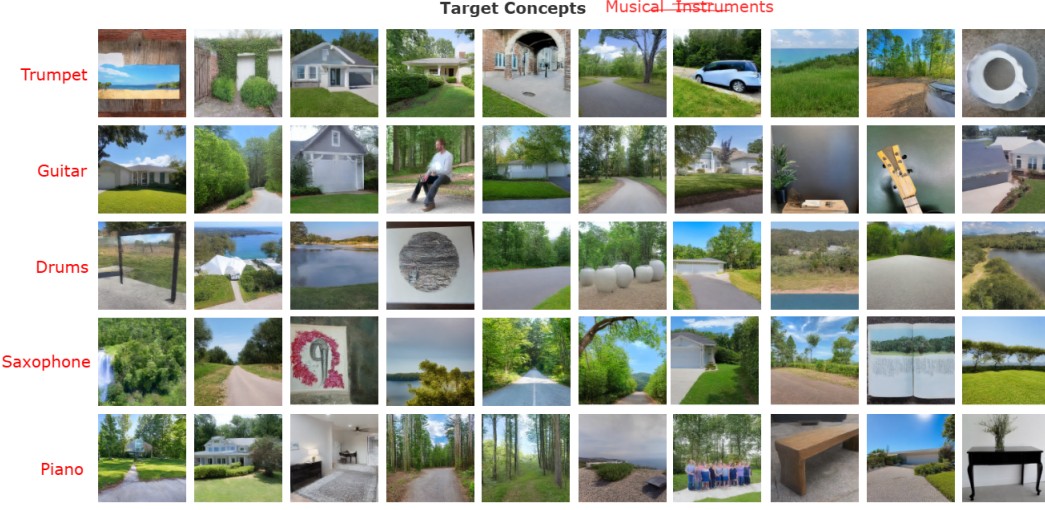

Figure 9: Illustration of the erasure of the *Musical Instruments* semantic category, along with visualizations of its subcategories (e.g., Trumpet, Guitar, Drums) generated by the erased model. The visual results clearly demonstrate that SURE effectively removes the entire semantic category.

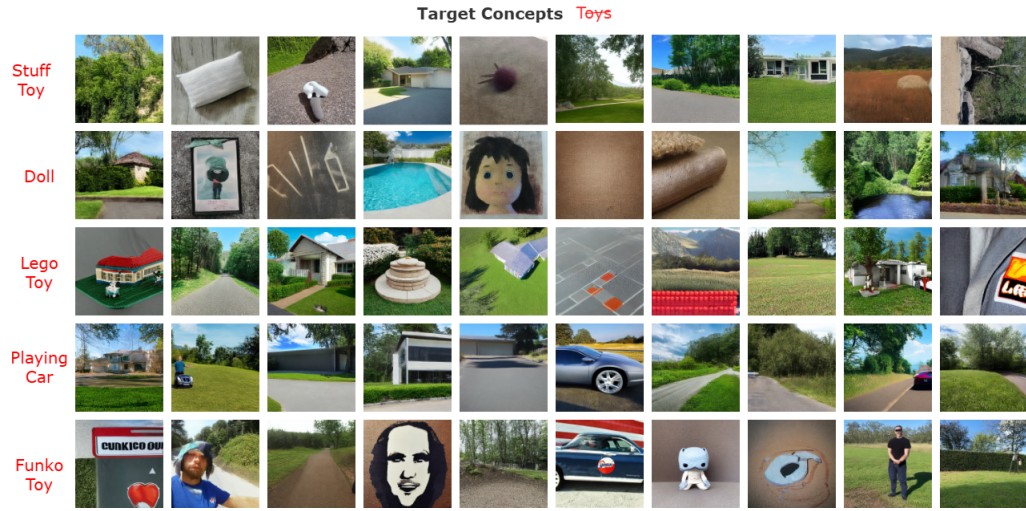

Figure 10: Illustration of the erasure of the *Toys* semantic category, along with visualizations of its subcategories (e.g., Stuff toy, Doll, Lego Toy) generated by the erased model. The visual results clearly demonstrate that SURE effectively removes the entire semantic category.

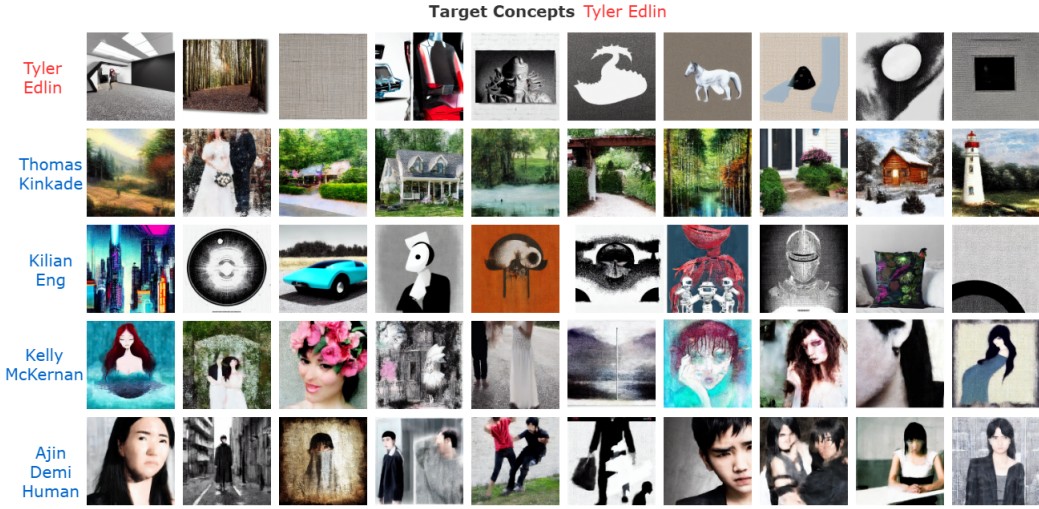

Figure 11: Illustration of the erasure of the artistic style of Tyler Edlin (shown in red), with preservation results displayed for the remaining artists (shown in blue).

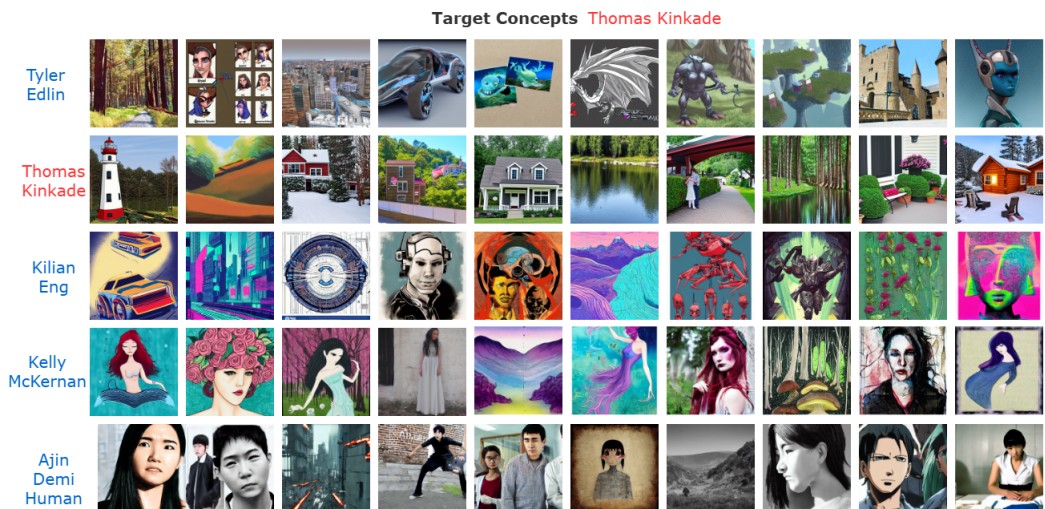

Figure 12: Illustration of the erasure of the artistic style of Thomas Kinkade (shown in red), with preservation results displayed for the remaining artists (shown in blue).

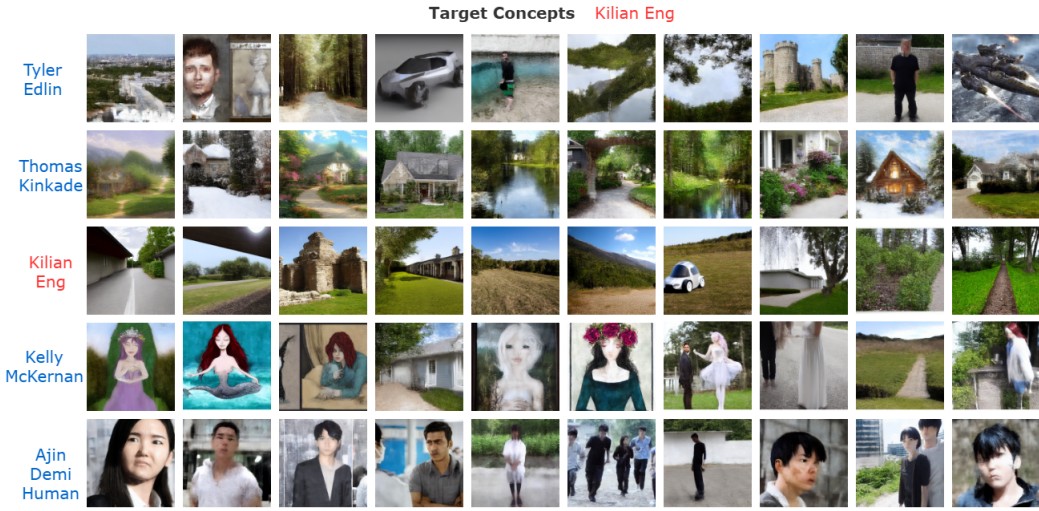

Figure 13: Illustration of the erasure of the artistic style of Kilian Eng (shown in red), with preservation results displayed for the remaining artists (shown in blue).

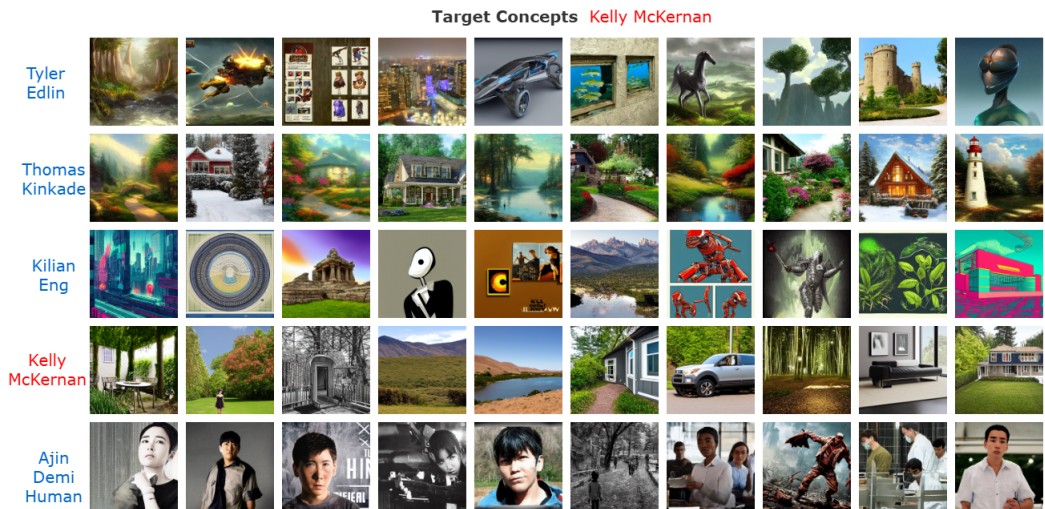

Figure 14: Illustration of the erasure of the artistic style of Kelly McKernan (shown in red), with preservation results displayed for the remaining artists (shown in blue).

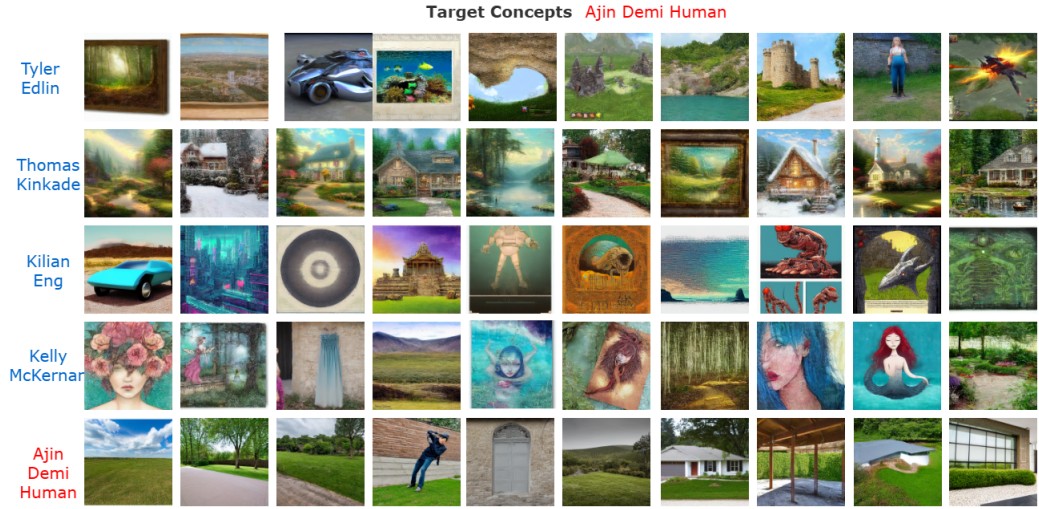

Figure 15: Illustration of the erasure of the artistic style of Ajin: Demi Human (shown in red), with preservation results displayed for the remaining artists (shown in blue).

