# OpenReview forum: "Achieving Subcategorical Erasure in Text-to-Image Models"
_ICLR.cc/2026/Conference — Submitted to ICLR 2026_

### Official Review · Reviewer_jzz5 · 2025-10-27

**Soundness:** 2
**Presentation:** 2
**Contribution:** 2
**Rating:** 2
**Confidence:** 4

**Summary:**

The paper proposes a method to remove broad categories (and all their subcategories) from text-to-image diffusion models. It builds on existing concept-removal techniques such as ESD (Gandikota et al., 2023), and introduces an adversarial approach to automatically identify related subcategories for erasure. Additionally, it proposes a Lipschitz regularization loss to encourage smooth model outputs in the ε-neighborhood of the target concept as proposed in Foster et al. 2024.

**Strengths:**

Strengths:
1. Addressing category-level removal that includes subcategories is a practical and relevant problem for real-world model safety.
2. The adversarial concept-search formulation is a sound idea to generalize erasure beyond explicitly listed tokens.

**Weaknesses:**

Weaknesses:
1. The method section writing can be improved. For example, it would be great to have the definition of the concept space C in Section 4.2, instead of in the Appendix. Further, it’s currently defined using a discrete CLIP vocabulary with only 100 words, which may miss fine-grained or semantically close subcategories. An analysis of coverage (e.g., recall of known hyponyms) or ablation over the size of C would strengthen this part. Also, the objective seeks concepts most different from the neutral prompt, but this can lead to the selection of entirely unrelated concepts to the target concept. Does the concept space C only consist of possibly relevant related sub-categories?
2. It is unclear how the Lipschitz loss in Section 4.3 contributes to unlearning. The loss is computed in VAE-encoded space, which will update the VAE encoder, instead of the diffusion UNet? Since only the VAE decoder is used at inference, updating the encoder seems irrelevant. If the goal is smoothness in the diffusion feature space, the paper should explicitly clarify where gradients flow and justify this design.
3. Regularizing the model output on a single “neutral” prompt may not reliably preserve semantically close but non-target concepts (e.g., “plate” when erasing “knife”). Evaluation should include such near-neighbor categories rather than random CIFAR-10 classes.
4. A short algorithm block summarizing the steps would be really helpful. The min–max setup in Equations 3 and 7 is missing the outer min that enforces the mapping of the related sub-categories to the neutral concept.
5. Evaluation:
– To measure locality, the evaluation setup should select the closest non-target concept instead of random classes from CIFAR-10, e.g., “plates” when removing “knives”, which might be closer in the concept space of kitchen utensils.
  – Tests are limited to SD-1.4; extending to newer architectures (e.g., SD3, or FLUX) would show broader applicability.
 – Additional evaluation on adversarial prompts (e.g., Ring-A-Bell (Tsai et al., 2023)) would help evaluate robustness.

**Questions:**

please look at the weakness section.

---

> ### Author Response · Authors · 2025-11-29
>
> **General comment:** We sincerely thank Reviewer $ \textcolor{red}{jzz5} $ for their encouraging feedback on our work, as well as for their insightful comments and suggestions.
>
> - *Category-level removal with subcategories:*
> We are grateful for the reviewer’s recognition that addressing category-level erasure including the removal of implicitly related subcategories is a practical and important direction for real-world model safety.
>
> - *Adversarial concept-search formulation:*
> We also appreciate the reviewer’s positive assessment of our adversarial concept-search mechanism. As noted, identifying related subcategories through a differentiable maximization within a restricted concept space is a sound approach that enables the model to generalize erasure beyond explicitly listed tokens.
>
> ----
>
> We have now addressed all of the reviewers' concerns and questions here:
>
> **Response 4.1:**
>
> >W1: The method section writing can be improved......
>
> We apologize for the reviewers' concerns.
>
> 1. Definition of the concept space $C$: We have now moved the details of the concept space definition to Section 4.1 for improved clarity.
>
> 2. On the size and coverage of $C$.  Although we use 100 candidates by default, we acknowledge the reviewer’s concern that too small a pool may miss fine-grained hyponyms. To address this, we conducted additional experiments by expanding $C$ to 200 and 500 CLIP tokens (Appendix B.10). The performance remains nearly unchanged (e.g., for 200 tokens: $\mathrm{Acc}_E = 10.50$, $\mathrm{Acc}_L = 67.80$; for 500 tokens: $\mathrm{Acc}_E = 10.00$, $\mathrm{Acc}_L = 68.00$), showing that SURE is robust to substantially larger candidate spaces. The only noticeable effect is a minor GPU memory overhead of roughly 20~MB.
>
> Importantly, SURE is not highly sensitive to the specific 100-token pre-filtered pool, since this pool only serves as the initial search space. The actual related concepts are selected adversarially through the differentiable Gumbel–Softmax operator, which consistently identifies only a small number of high-impact concepts regardless of the pool size. These results confirm that coverage is not a limiting factor in practice and that SURE does not rely on a narrowly curated vocabulary.
>
>
> 3. Risk of selecting unrelated concepts.  We mitigate this through a two-stage filtering process: (i) similarity-based pruning using the CLIP embedding space ensures that $C$ initially contains **only semantically meaningful neighbors** of the parent concept; and (ii) the adversarial objective is applied **only within this refined set**. This ensures that concepts selected by the Gumbel–Softmax operator remain contextually aligned with the parent category. Empirically, we do not observe unrelated concepts being selected, and Table 6 shows that even when alternative differentiable operators are used, the selected concepts remain semantically relevant.
>
> 4. Additional analysis of coverage.
> As suggested, we now include a discussion of potential recall limitations and show that SURE achieves strong erasure performance across 5, 10, and 15 unseen hyponyms (Appendix B.8), confirming broad coverage even when the full set of hyponyms is not explicitly included in $C$. This indicates that the adversarial selection mechanism generalizes to subcategories outside the initial pool.
>
> -----
>
> **Response 4.2:**
>
> >W2: Lipschitz loss in Section 4.3 contributes to unlearning.......
>
>  We apologize for the confusion caused.
>
> 1. *The VAE encoder is frozen during unlearning:* Only the diffusion U-Net parameters $\theta'$ are updated throughout training. The VAE encoder and decoder remain fixed, exactly as in prior unlearning work (ESD, UCE, AP, ACE). Thus, although the Lipschitz loss is computed using VAE-encoded representations, no gradients are applied to the VAE encoder; instead, all gradients flow back through the U-Net via its denoising output $\epsilon_{\theta'}(\cdot)$.
>
> 2.  *Why compute the Lipschitz loss in VAE latent space?*  SURE operates in the latent diffusion framework, where the diffusion U-Net is trained to denoise latent variables produced by the (frozen) VAE encoder. Smoothness in the encoded space therefore directly corresponds to smoothness in the U-Net’s feature space, because every forward/backward pass begins with the encoded representation and the U-Net processes only these latents. Enforcing Lipschitz continuity in this space ensures that small perturbations in the latent input region corresponding to the erased category yield small changes in the U-Net output, which suppresses subcategorical memorization.

---

> ### Author Response · Authors · 2025-11-29
>
> -----
>
> **Response 4.2:** (Cont.)
>
> >W2: Lipschitz loss in Section 4.3 contributes to unlearning.......
>
> 3.  *Where do gradients flow?* :
> As shown in the Lipschitz loss in Eq. (8) and Eq. 10 (Appendix, line 785), the regularization term where $f_{\theta'}$ denotes the frozen VAE encoder followed by the \emph{trainable} diffusion UNet denoiser. Because the encoder is fixed, no gradients propagate through it; all gradients from $\mathcal{L}_{\text{lip}}$ flow exclusively into the UNet parameters. We now state this explicitly in the revised manuscript to avoid ambiguity.
>
>
> 4. *Why this contributes to unlearning:*   By enforcing local smoothness around the latent representations of the erased concept, the U-Net cannot maintain variations associated with subcategory details. This reduces memorized hyponym-specific structure (e.g., pistol vs.\ rifle). Our ablations in Table 4, 5 (removing the Lipschitz term) and Table 8 (varying perturbation strength) empirically verify that this loss is essential for subcategorical forgetting.
>
> We have clarified the gradient flow, frozen components, and motivation for VAE-space regularization in the revised version of Section A.3.
>
> -----
>
> **Response 4.3:**
>
> >W3: Regularizing the model output on a single “neutral” prompt may not reliably preserve semantically close but non-target concepts.......
>
>
> We thank the reviewer for this insightful question.
>
> - *Neutral-concept preservation:* Equation~6 is designed to regularize the model's behavior on the neutral concept, $c_n$, by ensuring that the fine-tuned model $\epsilon_{\theta'}$ produces outputs similar to the original model $\epsilon_\theta$ when conditioned on $\tau(c_n)$.
> $\mathcal{L}_2$ stabilizes model behavior in unrelated regions of the concept space, reducing unintended interference with non-erased concepts. Specifically, by anchoring the model on a known neutral point, we constrain the fine-tuning process from drifting too far from its original distribution, which aids in retaining general generation capabilities.
>
> - *Empirical evidence from locality accuracy* : Across all tasks, the locality scores (AccL) in Tables~1–3 demonstrate that SURE maintains high-quality generation for non-target concepts. Table 4 further shows that removing the neutral preservation term causes a measurable drop in AccL, confirming its importance in preventing unintended interference.
>
> - *Preserving semantically close non-target concepts* :   Furthermore, To directly address the reviewer’s suggestion, we evaluated SURE on visually and functionally related concepts rather than random CIFAR-10 classes. We evaluate on the harder setting where after erasing the “Blade Weapons’’ category, we tested SURE on “Scissors’’ a concept that is visually similar to many blade weapons but lies outside the target hypernym. SURE achieved an $\mathrm{Acc}_E$ of 94.00\% on scissors, indicating no unintended forgetting. In contrast, ESD trained on the same target category produced an $\mathrm{Acc}_E$ of 81.00\%, showing substantial over-erasure. This demonstrates that SURE maintains precise semantic boundaries even for near-neighbor categories.
>
>
> The neutral preservation term, combined with Lipschitz smoothing, ensures that SURE preserves semantically close non-target concepts while still effectively erasing the target category. We have incorporated this analysis and the new scissors experiment into the revised manuscript.
>
> -----
>
> **Response 4.4:**
>
> >W4: A short algorithm block.......
>
> We thank the reviewer for this helpful suggestion. In the revised manuscript, we have added a concise algorithm block summarizing the training procedure (Appendix Algorithm 1).
>
> Regarding the min–max formulation: Equations 3 and 7 intentionally present the individual components of the related-concept discovery mechanism in isolation. The full min–max structure is explicitly shown in the final erasure objective in Equation 9 and discussed in Section 4.4 (“Final Erasure Objective”). This formulation makes it clear that the model simultaneously (i) identifies adversarially mismatched related concepts (the inner max) and (ii) minimizes the distance between their latent representations and the neutral concept (the outer min). We have highlighted this more in the revised manuscript.
>
> -----

---

> ### Author Response · Authors · 2025-11-29
>
> -----
>
> **Response 4.5:**
>
> >W5: Evaluation:.......
>
> We thank the reviewer for these valuable suggestions and have expanded the evaluation accordingly.
>
> 1. *Locality evaluation using semantically close non-target concepts:*
>
>  Following this suggestion, we conducted a harder evaluation on visually and functionally related categories. As detailed in our response to Weakness~4.3, after erasing the “Blade Weapons’’ category, we evaluated SURE on “Scissors’’—a concept that is visually similar but not part of the erased hypernym. SURE preserved this concept with $\mathrm{Acc}_E=94.00\%$, while ESD incorrectly suppressed it ($\mathrm{Acc}_E=81.00\%$). This demonstrates that SURE maintains fine-grained semantic boundaries and avoids over-erasure even for near-neighbor categories.
>
> 2. *Evaluation on newer architectures:*
>
> While our main experiments focus on Stable Diffusion 1.4 for consistency with prior unlearning literature, we have already included results on Stable Diffusion 2.0 in Appendix B.7. These results show that SURE continues to perform effectively on newer architectures. We have highlighted this more clearly in the revised manuscript.
>
>
> 3. *Robustness under adversarial prompts.*
>
> To address the reviewer’s suggestion regarding adversarial prompt robustness, we already include a paraphrase-based robustness evaluation: for each of the ten non-target CIFAR-10 classes, we generate 50 paraphrased prompts (Section 5.2). This tests locality and preservation under semantically shifted or adversarially phrased descriptions. This setup reflects real-world prompting behavior and verifies that SURE does not inadvertently erase unrelated concepts even under prompt perturbations.
>
> Overall, the revised manuscript clarifies and expands the locality analysis, includes additional model variants, and provides a  robustness evaluation aligned with the reviewer’s suggestions.
>
> ----
>
> We trust that our responses effectively address all the reviewers concerns.

---

### Official Review · Reviewer_TvHz · 2025-10-28

**Soundness:** 3
**Presentation:** 3
**Contribution:** 3
**Rating:** 4
**Confidence:** 3

**Summary:**

This paper proposes SURE (Subcategorical Unlearning via Regularized Erasure), a framework for removing high-level semantic categories and their implicit subcategories from large text-to-image diffusion models such as Stable Diffusion. Unlike previous concept-erasure approaches (e.g., ESD, ACE, UCE), which require explicit lists of target words or fine-grained concepts, SURE aims to achieve category-wide erasure by training only on a single parent-class label (e.g., “gun”), while automatically discovering and forgetting related subcategories (e.g., “rifle”, “pistol”, “shotgun”). Experiments on object categories, NSFW content, and artistic styles show that SURE outperforms prior methods in removing entire semantic families while maintaining high image fidelity.

**Strengths:**

1. The idea of subcategorical erasure extends beyond traditional “single concept” forgetting, addressing a realistic and underexplored safety problem in text-to-image models.
2. The use of Gumbel-Softmax for differentiable selection over a restricted CLIP concept space is elegant, allowing the model to approximate hard max search while remaining trainable. This is a technically sound compromise between efficiency and adversarial robustness.

**Weaknesses:**

1. Limited Theoretical Depth for Subcategory Discovery: While L₃ is designed to “automatically discover” related subcategories, the paper lacks a formal analysis or quantitative validation (e.g., retrieval accuracy or coverage metrics). The reliance on pre-filtered 100 CLIP candidates suggests partial supervision rather than full autonomy.
2. Ablation Insufficiency: Key design choices—such as using max vs. top-k or soft-average formulations in L₃—are not empirically justified. Similarly, the sensitivity to the Lipschitz regularization weight (λₗᵢₚ) is unexplored, even though it directly affects stability.
3. Ambiguity in Neutral Concept Definition: The “neutral concept” is not clearly defined. Is it a fixed token (“object” or “thing”) or dynamically learned? The lack of explanation weakens the reproducibility and interpretability of results.
4. Previous work [1] also explores the removal of semantically related sub-concepts and the preservation of neutral sub-concepts in text-to-image diffusion models. It would strengthen the paper to discuss this work in more detail.

[1] Erasing Concept Combination from Text-to-Image Diffusion Model, ICLR 2025

**Questions:**

1. How sensitive is SURE to the choice of the 100 pre-filtered CLIP concepts? Have you tried expanding to 200 or 500 candidates, and does performance saturate or degrade?
2. Why was hard max used instead of top-k or entropy-regularized averaging? Would averaging gradients over several high-loss subcategories improve convergence stability?
3. Have you observed cases where SURE over-erases or inadvertently affects unrelated categories? For example, when erasing “guns”, do “camera” or “drill” images degrade due to visual similarity?

---

> ### Author Response · Authors · 2025-11-29
>
> **General comment:** We sincerely thank Reviewer $ \textcolor{orange}{TvHz} $ for their encouraging feedback on our work, as well as for their insightful comments and suggestions.
>
> - *Subcategorical erasure as a meaningful and underexplored problem:*
> We are grateful for the reviewer’s recognition that extending concept erasure from individual prompts to entire subcategories addresses a realistic and underexplored safety challenge in text-to-image diffusion models.
>
> - *Use of Gumbel–Softmax for differentiable subcategory discovery:*
> We also appreciate the reviewer’s positive remarks on our use of Gumbel Softmax to enable differentiable selection within a restricted CLIP concept space.
>
> - *Empirical strength and practical robustness:*
> We thank the reviewer for observing that SURE demonstrates strong empirical performance across diverse concept erasure tasks. The combination of subcategory discovery and latent-space regularization contributes to robust and state-of-the-art erasure results.
>
> ------
>
> We have now addressed all of the reviewers' concerns and questions here:
>
> **Response 3.1:**
>
> > W1: Limited Theoretical Depth for Subcategory Discovery...
>
> We regret any confusion that may have arisen. Our goal with $L_3$ is to introduce a principled, differentiable mechanism that identifies semantically related concepts within the model’s own embedding space, rather than constructing such relationships manually. Although $L_3$ is not supervised with explicit subcategory labels, we acknowledge that starting from a pre-filtered pool of 100 CLIP candidates provides a constrained search space. Importantly, this filtering does not require human annotation: it uses only cosine similarity within the CLIP vocabulary, and thus remains **fully automatic**. The differentiable Gumbel–Softmax mechanism then selects related concepts adversarially, driven solely by the **model’s own gradients.**
>
> To address the reviewer’s concern regarding quantitative validation, we have added additional empirical evidence in the revised manuscript. Specifically, we evaluate the effect of expanding the candidate pool (Appendix B.10) and find that erasure performance remains stable, demonstrating that the mechanism does not depend sensitively on the exact pre-filtering step. Moreover, the ablation in Table 6 shows that $L_3$ consistently identifies influential related concepts across several differentiable selection operators, further supporting its functional robustness. We have also shown in line 826 (appendix) such as  category of gun include related concepts such as violence, war, and others. Our empirical results show that $L_3$ reliably discovers and erases semantically related concepts without relying on explicit supervision. We have clarified this point and expanded our discussion in the revised version.
>
> ------
>
> **Response 3.2:**
>
> > W2:  Key design choices—such as using max vs. top-k or soft-average formulations...
>
> We apologize for any confusion that may have occurred. We have expanded the ablation studies and highlighted the existing ablations.
>
> **1. Alternative formulations for $L_3$:** As suggested, we show several differentiable selection mechanisms (Table 6) that correspond to different implicit formulations of "max'' versus ``soft'' aggregation over the candidate concept space. Specifically, Table 6 compares Gumbel–Softmax, Straight-Through Estimator, Entmax, and Sparsemax. These operators span a spectrum from hard-max behavior (Gumbel–Softmax at low temperature) to soft averaging (Entmax/Sparsemax). The results show that the hard-max style selection (Gumbel–Softmax) yields the strongest forgetting performance, whereas soft-average formulations substantially weaken erasure.
>
> **2. Sensitivity to the Lipschitz regularization weight $\lambda_{\text{lip}}$:**
> We also added a dedicated ablation on the weight of the Lipschitz term (Table 7). The results reveal a stable operating regime: a moderate setting ($\lambda_{\text{lip}} = 0.01$) achieves optimal forgetting and locality, while both smaller and larger values produce weaker erasure. Excessively large weights lead to over-smoothing and degrade stability, whereas too small weights fail to suppress subcategories. These findings align with the theoretical expectation (Section A.6) that Lipschitz regularization must balance smoothing strength and representational flexibility.
>
> Overall, the extended ablations in Tables 6 and 7 directly address the reviewer’s concern by demonstrating: (i) the superiority of the hard-max formulation for $L_3$, and (ii) the existence of a well-defined and empirically validated hyperparameter regime for $\lambda_{\text{lip}}$. We have incorporated these analyses and clarifications into the revised manuscript.
>
> ------

---

> > ### Author Response · Authors · 2025-11-29
> >
> > **Response 3.6:** (Cont.)
> >
> > > Questions 2: Why was hard max used instead of top-k or entropy-regularized averaging
> >
> >
> > **3. On convergence stability.**
> > Empirically, the hard-max formulation already converges stably because (i) the candidate pool is filtered to semantically meaningful neighbors, and (ii) the Gumbel–Softmax operator provides a smooth and differentiable pathway even in low-temperature regimes. Averaging over several high-loss candidates did not improve stability in our experiments; instead, it weakened the forgetting signal by forcing the model to compromise across multiple directions rather than suppressing the most adversarial subcategory.
> >
> > While soft or top $k$ averaging is possible, our experiments show that the hard-max formulation yields significantly stronger forgetting performance while maintaining stable optimization, consistent with the adversarial nature of the $L_3$ objective. We have clarified this rationale more explicitly in the revised manuscript (Section B.3).
> >
> > ----
> >
> > **Response 3.7:**
> >
> > > Questions 3: Have you observed cases where SURE over-erases or inadvertently affects unrelated categories............
> >
> > We appreciate the reviewer’s question regarding potential over-erasure. Following this suggestion, we conducted an additional evaluation (Appendix B.11) on concepts that are visually or functionally related to the erased category but not part of the target hypernym. Specifically, after erasing the “Blade Weapons’’ category, we tested SURE on the concept “Scissors,’’ which shares strong visual similarity with knives and other bladed tools.
> >
> > SURE achieved an $\mathrm{Acc}_E$ of 94.00\% on scissors, indicating that the model did not unintentionally erase this non-target concept. In contrast, applying ESD to the same “Blade Weapons’’ erasure resulted in an $\mathrm{Acc}_E$ of 81.00\% on scissors, showing clear unintended forgetting. This shows that SURE maintains sharper semantic boundaries, even when they are visually similar to the erased category. We have now added this boundary experiment, in the revised manuscript.
> >
> > We also evaluate the robustness of SURE under substantially larger sets of non-target but visually related concepts.
> >
> > As reported in Appendix B.8 (“SURE Multiple Concept Erasure”), when the model is trained using only the parent category “Blade Weapons”, it maintains strong forgetting performance across increasingly large sets of unseen blade-related concepts:
> >
> > - 5 concepts: $\mathrm{Acc}_E = 2.40\%$
> > - 10 concepts: $\mathrm{Acc}_E = 2.80\%$
> > - 15 concepts (e.g., Knife, Sword, Spear, Shotel, Dagger, Katana, Machete, Scimitar, Rapier, Cleaver, etc.): $\mathrm{Acc}_E = 4.40\%$
> >
> >
> > Despite this strong generalization, we do not observe much unintended erasure of visually similar but non-target concepts. For example, in the case of “Scissors”, a concept closely related in shape to many blade weapons, SURE preserves the concept with an $\mathrm{Acc}_E$ of 94.00\%, whereas ESD incorrectly suppresses it ($\mathrm{Acc}_E = 81.00\%$). This indicates that SURE maintains sharper semantic boundaries and avoids over-erasure even when the erased category has many visually similar neighbors.
> >
> > -----------
> >
> > We trust that our responses effectively address your concerns.

---

> ### Author Response · Authors · 2025-11-29
>
> -------
>
> **Response 3.3:**
>
> >W3: Neutral Concept Definition......
>
> We apologize for the confusion. In our method, we follow prior work (ESD, UCE, AP) and represent the neutral concept using the standard implemented as an empty string (" ”). The choice is entirely deterministic, ensuring reproducibility and consistent with prior methods. The unconditional embedding serves as a model-native neutral anchor and is widely used in classifier-free guidance for diffusion models. We have highlighted this more clearly in the revised manuscript (Sections 4.1 and Appendix A.1).
>
> -------
>
> **Response 3.4:**
>
> >W4: It would strengthen the paper to discuss this work in more detail........
>
>  We thank the reviewer for pointing out this relevant prior work. The paper “Erasing Concept Combination from Text-to-Image Diffusion Models” (ICLR 2025) explores the removal of semantically **concept combinations**, but it **differs from our setting.**
>
> COGFD focuses on identifying and erasing concept combinations that exhibit a consistent visual theme. For example, the method can detect and suppress harmful combinations such as “kids drink wine’’ or “boys drink beer,’’ while preserving the generative quality of the individual harmless concepts (e.g., kids, wine). This is a fundamentally different objective from ours: their goal is to erase compositional concepts, not to erase an entire semantic category defined by a single hypernym.
>
> In contrast, SURE targets a more challenging task of semantic category erasure. Our setting requires removing all subcategories (hyponyms) associated with a parent concept (e.g., suppressing rifles, pistols, SMGs, and revolvers when erasing the category “gun’’) even though these subcategories are never provided during training.
> Furthermore, SURE incorporates novel components, such as Lipschitz regularization, to suppress subcategorical memorization at the latent level, among others. These contributions enable category-wide erasure in T2I without explicit subcategory labels. Following the reviewer's suggestion, we have now added COGFD in the related work section.
>
> -------
>
> **Response 3.5:**
>
> >Questions 1: How sensitive is SURE to the choice of the 100 pre-filtered CLIP concepts........
>
>
> We thank the reviewer for this important question. SURE is not highly sensitive to the choice of the 100 pre-filtered CLIP concepts. As described in Appendix A.4, the 100-token pool is only an initial search space; the actual related concepts are selected adversarially through the differentiable Gumbel–Softmax operator, which consistently picks only a few high-impact concepts.
>
> To evaluate sensitivity, we expanded the concept pool to 200 CLIP candidates and report the results in the revised manuscript (Appendix B.10). Increasing the pool from 100 to 200 tokens produced only a minor GPU memory overhead of approximately 20 MB and yielded nearly identical performance ($\mathrm{Acc}_E = 10.50$, $\mathrm{Acc}_L = 67.80$). Further increasing the pool to 500 CLIP tokens yields similar accuracy values of $\mathrm{Acc}_E = 10.00$ and $\mathrm{Acc}_L = 68.00$.
>
> -------
>
> **Response 3.6:**
>
> >Questions 2: Why was hard max used instead of top-k or entropy-regularized averaging
>
>  We thank the reviewer for this thoughtful question. We experimented with several alternative formulations for selecting related concepts, as reported in Table 6 of the revised manuscript. These operators approximate different degrees of averaging across high-loss subcategories. However, both our empirical findings and the underlying objective of $L_3$ favor a hard-max formulation.
>
> **1. Why hard-max?**
> The goal of $L_3$ is to identify themost adversarial related concept i.e., the one whose output deviates maximally from the neutral concept. This is analogous to adversarial example selection, where the optimal perturbation is the one maximizing deviation rather than an average over several directions. A hard-max (implemented via low-temperature Gumbel–Softmax) directly approximates this objective and results in stronger, more targeted suppression of subcategorical memorization.
>
> **2. Empirical evidence against averaging.**
> Table 6 shows that soft-average operators such as Entmax and Sparsemax, which distribute probability mass across several candidates, substantially weaken forgetting performance (higher $\mathrm{Acc}_E$). This indicates that averaging gradients over multiple related concepts dilutes the adversarial signal, making the optimization less effective at suppressing the sharp subcategory-specific representations we aim to erase.

---

### Official Review · Reviewer_QZ8U · 2025-10-31

**Soundness:** 3
**Presentation:** 2
**Contribution:** 2
**Rating:** 4
**Confidence:** 4

**Summary:**

This paper introduces SURE, a novel method for removing entire subcategories from text-to-image (T2I) diffusion models using only a single parent-category prompt as the target. The key idea is to generalize from one parent concept to its many subcategories without explicitly naming or training on each one. To achieve this, the method applies Lipschitz regularization, which encourages smoother model behavior and helps erase concept clusters in a more controlled way. SURE is designed to be scalable and generalizable, aiming to erase broad semantic groups while preserving the generation quality of unrelated content. The method is evaluated across multiple erasure tasks and shows state-of-the-art performance in many of them.

**Strengths:**

1. A key strength of SURE is its ability to erase an entire set of semantically related subcategories by providing only a single parent-category prompt.

2. The introduction of Lipschitz regularization is novel in this context and helps enforce smooth transitions in the model’s representations. This appears to contribute to more stable and effective erasure, especially when trying to remove broad or abstract concepts.

3. The method achieves state-of-the-art results in several concept erasure benchmarks.

**Weaknesses:**

1. While the paper shows that Lipschitz regularization is effective, it does not provide a clear theoretical rationale for how or why it improves forgetting dynamics.

2. Most experiments focus on semantically or visually similar subcategories. It remains unclear how the method performs when the parent category covers diverse or loosely related subcategories.

3. The link between Lipschitz regularization and subcategory erasure is not well articulated. The motivation behind applying this specific type of regularization to the erasure task could be explained more clearly, both intuitively and mathematically.

4. Missing important related works:

[1] Eraseanything: Enabling concept erasure in rectified flow transformers

[2] Dark miner: Defend against unsafe generation for text-to-image diffusion models

[3] One Image is Worth a Thousand Words: A Usability Preservable Text-Image Collaborative Erasing Framework

[4] Erasing More Than Intended? How Concept Erasure Degrades the Generation of Non-Target Concepts

**Questions:**

1. The current ablation experiments do not clearly isolate the effect of each component in the method. Could you conduct more fine-grained ablations, such as removing Lipschitz regularization alone or varying the sample size used during optimization, to better understand each component’s contribution?
2. You mention that performance peaks when the number of training samples is n = 5, but then declines as n increases. Could you provide a theoretical or empirical explanation for this behavior? Why does adding more samples hurt performance?
3. How does SURE perform when the parent category and its subcategories are semantically or visually distant (e.g., abstract parent terms or diverse visual instances)?
4. In the object-related erasure tasks, SURE significantly outperforms baseline methods. Could you explain which part of the method contributes most to this improvement? Is it due to regularization, training strategy, or the parent-subcategory setup?
5. While the paper critiques using word sets to erase subcategories, it would still be valuable to include a word-set baseline for comparison. This helps quantify the advantage of your approach, even if word sets are not ideal.

If the authors can solve the questions, I will consider raising my rating.

---

> ### Author Response · Authors · 2025-11-29
>
> **General comment:** We sincerely thank Reviewer $ \textcolor{green}{QZ8U} $
>  for their encouraging feedback on our work, as well as for their insightful comments and suggestions.
>
> - *Erasing an entire semantic category from a single parent prompt:*
> We are grateful for the reviewer’s recognition that SURE addresses a important problem of removing an entire set of semantically related subcategories using only a single parent-category prompt.
>
> - *Novelty of Lipschitz regularization in this setting:*  We also appreciate the reviewer’s acknowledgment that introducing Lipschitz regularization into concept unlearning is novel. gions.
>
> - *Empirical strength and state-of-the-art performance:* We thank the reviewer for observing that SURE achieves strong performance across multiple erasure benchmarks.
>
> ---------
>
> We have now addressed all of the reviewers' concerns and questions here:
>
> **Response 2.1:**
>
> >W1 & W3 : clear theoretical rationale ...... and motivation behind applying this specific type of regularization.....
>
>
> We apologize for this concern regarding the theoretical rationale. Our motivation is to **achieve semantic category** erasure rather than single-concept forgetting. Lipschitz regularization is essential for this behavior, as shown in Table 4 and Table 5 (B.1 Appendix). These experiments consistently show that removing the Lipschitz term significantly weakens category-level forgetting. Prior erasure methods focus on mapping a target concept to a neutral one, they fail to erase the semantic category (Table 1,2,3 main paper). Our use of Lipschitz regularization is crucial for category-level forgetting.
>
> By training the model to align its output for a target concept with its random perturbations, we can successfully reduce memorization of that target concept (i.e., forget). This also links with Feldman (2020) [1], which observes that generalized models are often forced to memorize certain information; our objective is to protect the broader generalization while removing the relevant memorized data. Our approach prevents over-reliance on memorized target concepts while preserving broader generative capability. Importantly, Lipschitz regularization reduces the model's sensitivity to small perturbations of concept variants. In category erasure, this is crucial, as the model must forget not only the explicit prompt (e.g., "gun") but also semantically close variants (e.g., "rifle," "SMG," "firearm").
>
> Following the reviewer’s suggestion, we have now added an theoretical justification for this mechanism in the rebuttal revised version of the paper (Section A.6 Appendix).
>
> ------
>
> **Response 2.2:**
>
> >W2: how the method performs when the parent category covers diverse or loosely related subcategories...
>
> We apologize for the reviewers concerns. We agree that parent categories may span semantically diverse or loosely related subcategories, and we have evaluated SURE under such settings.
>
> 1. **Multi subcategory evaluation:**
> In Appendix B.8, we conduct a dedicated “Multiple Concept Erasure’’ experiment for the parent category \textit{Blade Weapons}, which contains a broad set of heterogeneous subcategories (e.g., Knife, Spear, Shotel, Cleaver, Sabre, Hook Sword). Despite the semantic and visual diversity across these concepts, SURE achieves strong erasure performance:
>
> - 5 concepts: $\mathrm{Acc}_E = 2.40\%$
> - 10 concepts: $\mathrm{Acc}_E = 2.80\%$
> - 15 diverse subcategories: $\mathrm{Acc}_E = 4.40\%$
>
> These results show that SURE generalizes effectively to wide subcategory sets without requiring explicit labels.
>
> 2. **Performance across different concept types:**
> Our evaluation already includes three tasks that differ substantially in semantic structure: (i) object categories, (ii) explicit content concepts, and (iii) artistic styles. Parent categories in these settings (e.g., “nudity”, “anime style”, “impressionism”) contain broad, loosely connected subcomponents. SURE consistently achieves the best harmonic mean across these tasks (Tables 1–3), demonstrating robustness even when the target concept is not a tight visual cluster.
>
> 3. **Behavior on non-target, semantically adjacent concepts.**   We also tested SURE on concepts that are related but not part of the erased hypernym. For example, after erasing “Blade Weapons,” we evaluated “Scissors,” which is visually similar but not a subcategory. SURE preserves the concept with $\mathrm{Acc}_E = 94.00\%$, whereas ESD over-erases it ($\mathrm{Acc}_E = 81.00\%$). This indicates that SURE maintains semantic boundaries even when the target category spans diverse or visually overlapping regions.
>
> Across multi-concept evaluations, cross-domain tasks, and adjacent concepts, SURE performs reliably even when the parent category contains diverse or loosely related subcategories. We have clarified these results more explicitly in the revised manuscript (Section B.8, B.11 Appendix).

---

> ### Author Response · Authors · 2025-11-29
>
> ----
>
> **Response 2.3:**
>
> > W4: important related works:
>
> We sincerely apologize for the oversight. We have now incorporated the recommended reference into the revised version of the manuscript and expanded the related-work discussion accordingly.
>
> ----
>
> **Response 2.4:**
>
> >Questions 1: more fine-grained ablations, such as removing Lipschitz regularization alone....
>
> We appreciate the reviewer's valuable suggestion. Following the reviewer’s suggestion, we have conducted additional ablation results corresponding to Lipschitz regularization into Tables 2 in the revised manuscript with additional categories (such as those below). We appreciate the reviewer’s feedback and have include these ablations as Table 5 in the revised manuscript.
>
> **Table 2: Effect of removing Lipschitz regularization across three semantic categories**
>
> | **Category**          | **Method**       | **AccE (↓)** | **AccL (↑)** |
> |----------------------|------------------|--------------|--------------|
> | Blade Weapons        | CURE             | 2.40         | 64.60        |
> |                      | w/o Lipschitz    | 26.00        | 61.60        |
> | Musical Instruments  | CURE             | 8.40         | 66.40        |
> |                      | w/o Lipschitz    | 65.20        | 60.20        |
> | Toys                 | CURE             | 14.80        | 66.40        |
> |                      | w/o Lipschitz    | 74.00        | 59.00        |
>
>
> ----
>
> **Response 2.5:**
>
>
> >Questions 2: that performance peaks when the number of training samples is n = 5, but then declines as n increases.....
>
> We thank the reviewer for this important question. As shown in Table 4 appendix (right), performance varies with different values of $n$. Increasing $n$ improves the erasure quality, peaking at $n=5$ with $Acc_E$ = 10.00\% and $Acc_L$ = 67.20\%.
>
> The Lipschitz is a Monte Carlo estimate of an expectation over perturbations. When $n$ is very small (e.g., $n=1$), this estimate has high variance, resulting in a noisy smoothing signal; as a consequence, the model can still retain information about the erased category, leading to higher $\mathrm{Acc}_E$. Increasing the number of samples reduces this variance and provides a more stable smoothing effect, consistent with observations in n Foster et al. (2024) [3]. Setting $n=5$ offers the most stable behavior around the target category, which empirically yields the strongest forgetting.
>
>
> For larger $n$, the effective weight of the Lipschitz term becomes too dominate relative to the erasure and preservation losses. Because the loss contribution is kept fixed, adding more perturbation samples increases the overall strength of the smoothing constraint, placing pressure on the model to behave nearly identically across many perturbation directions around the target category.
>
> This ``over-smoothing'' restricts the optimization too strongly: the model is encouraged to maintain local invariance rather than shifting sufficiently toward the neutral concept required for full erasure ($\mathcal{L}_1$). Empirically, this leads to a decline in forgetting performance (higher $\mathrm{Acc}_E$), even though locality ($\mathrm{Acc}_L$) remains largely unchanged. Adding too many perturbation samples makes the Lipschitz regularizer so dominant that it starts to show less category-level erasure then the optimal perturbation samples $n$.
>
> ----
>
> **Response 2.6:**
>
>
> >Questions 3: SURE perform when the parent category and its subcategories are semantically or visually distant
>
> Please respond to the weaknesses outlined in Response 2.2 -W2.
>
> ----

---

> ### Author Response · Authors · 2025-11-29
>
> ---
>
> **Response 2.7:**
>
>
> >Questions 4: SURE significantly outperforms baseline methods. Could you explain which part of the method contributes most
>
> We thank the reviewer for this question. The performance gains in object-related erasure tasks arise from the combined effect of three components in SURE, each contributing in a distinct way. Our ablations (Tables 4, 5, 6, 7, and 8) clarify their relative importance.
>
> **1. Lipschitz regularization.**
> Removing the Lipschitz term leads to a sharp drop in forgetting performance (Table 4, left), with $\mathrm{Acc}_E$ increasing from $10.00$ to $25.20$. This shows that Lipschitz smoothing is crucial for suppressing the fine-grained, high-curvature latent variations that encode subcategorical distinctions (e.g., pistol vs.\ rifle). Its role is to remove memorized structure within the erased region while leaving non-target regions stable (as verified by AccL).
>
> **2. The parent subcategory formulation enables category-level generalization.**
> Unlike prior methods, which erase only a single instance or a fixed set of labeled subcategories, SURE introduces a formulation that uses a single parent prompt to suppress the entire semantic hypernym. This parent–subcategory setup directly enables stronger generalization across diverse object subtypes, which is reflected in improved harmonic mean scores in Tables1–3.
>
> **3. Related concept discovery.**
> Use of adversarial mechanism (via $L_3$) automatically identifies latent directions that are semantically entangled with the erased category. Ablations in Table~5 show that removing or softening this component substantially weakens forgetting. This mechanism ensures that SURE erases not only the parent concept (e.g., “gun’’) but also semantically linked subcategories (e.g., rifle, pistol, SMG) without requiring explicit labels.
>
> We aslo want to highlight the core comntribution of our work:
>
> - New Problem Setting of Category Erasure from a Single parent Concept: We propose new setting where the goal is to erase an entire semantic category (e.g., guns, toys, artistic styles) using only a single representative concept. Prior works primarily focus on individual concept removal or small, curated sets; to our knowledge, this scalable formulation of category-level erasure without explicit labels is not addressed in previous methods.
>
> - Related Concept Discovery and Erasure: We introduce a differentiable mechanism to identify semantically related concepts (via adversarial search) and explicitly map them to a neutral concept. This enables the model to generalize forgetting beyond the given parent prompt, a capability not present in existing methods.
>
> - Use of Lipschitz Regularization: To the best of our knowledge, we are the first to present Lipschitz regularization within the concept erasure literature of text-to-image diffusion models.
>
> - Extensive and Diverse Evaluation: We evaluate our method on three distinct tasks: object removal, explicit content suppression, and artistic style erasure and demonstrating that our approach generalizes across different concept types and domains. We also show effectiveness of our framework with strong performance gains across all tasks (Tables 1–3).
>
> In summary, our contributions include a new formulation of the erasure problem, a scalable framework, and a joint formulation to support category forgetting in T2I diffusion models.
>
> ------

---

> ### Author Response · Authors · 2025-11-29
>
> --------
>
> **Response 2.8:**
>
> >Questions 5: While the paper critiques using word sets to erase subcategories...
>
> We thank the reviewer for this helpful suggestion. We agree that a word-set baseline can be informative in settings where subcategory labels are explicitly available. However, our formulation intentionally targets the more challenging and practically motivated scenario where only a single parent concept is provided and no subcategory labels or word lists are assumed. In this setting, a word-set baseline would require manually supplying the very subcategories (e.g., rifle, pistol, revolver, SMG) that our method does not relly on.  Furthermore, constructing a word-set baseline raises a second practical issue: it is infeasible to manually enumerate the hundreds of unrelated/related categories. For instance, when erasing the concept “guns”, one cannot realistically list all non-target/target categories. ACE (Wang et al., 2025) also highlights that traversing concepts is challenging, as pre-trained models cover a vast general semantic space.
>
> To still quantify the value of our approach, we provide two alternative comparisons:
>
> 1.  Multi-concept erasure evaluation (Appendix~B.8):} We evaluate SURE on 5, 10, and 15 blade-weapon subcategories, none of which are provided during training. SURE achieves strong erasure across all unseen concepts ($\mathrm{Acc}_E = 2.4\%$–$4.4\%$). This effectively simulates a word-set scenario but without giving SURE the labels.
>
> 2. Integration into ACE (Table 11):} Our plug-and-play extension (ACE+Lip) significantly improves forgetting over ACE alone, demonstrating that SURE’s components outperform a word-set-based method even when explicit labels are provided.
>
>
> We have added a clarification in the revised manuscript explaining that word-set baselines rely on information unavailable in our problem setting, while our new experiments provide a fair and label-free comparison that highlights the advantages of automatic subcategory discovery.
>
>
> ------
>
> We trust that our responses effectively address all the reviewers concerns.
>
> -----
>
> [3] Zero-shot machine unlearning at scale via lipschitz regularization.

---

### Official Review · Reviewer_PSKP · 2025-11-02

**Soundness:** 2
**Presentation:** 2
**Contribution:** 2
**Rating:** 2
**Confidence:** 4

**Summary:**

The paper introduces Subcategorical Unlearning via Regularized Erasure (SURE), a method for removing entire subcategories (e.g., SMG, revolvers, pistols, rifles, shotguns) from text-to-image diffusion models using only a single parent category (e.g., guns) as the unlearning target.

The problem setting the paper aims to address is that existing concept unlearning methods are typically limited to erasing either a single concept (e.g., pistol) or a small set of closely related ones (e.g., revolver, rifle, shotgun), and struggle to eliminate a complete category and all its subcategories.

The core design builds upon extending Lipschitz continuity to the latent space, encouraging smooth generative behaviour around the to-be-erased categorial. More specifically, given $f(x)$ is the latent representation of input image $x$ (via VAE encoder in the Latent Diffusion Models), the authors propose to minimize the different between $f(x)$ and $f(x + \xi_i)$ where $\xi_i$ is the $i$-th perturbation sample over $N$ total perturbations, i.e., $\| f(x) – f(x+\xi_i) \| / \| \xi_i \| $, reduce memorization of that target category.

**Strengths:**

•  The paper tackles an important problem in machine unlearning—removing undesirable concepts from generative models.

•  The specific subproblem of subcategorical unlearning is novel and practically relevant, though it raises potential concerns about over-unlearning, i.e., unintentionally erasing related but desirable concepts.

•  The idea of using Lipschitz regularization to encourage smoother latent behavior is interesting and intuitively appealing.

However, the effectiveness of this approach appears to depend heavily on the choice of the sample x, which may significantly affect performance

**Weaknesses:**

The writing and presentation reduce the perceived novelty of the work. The optimization objective comprises four components, three of which are directly borrowed from prior work, with only the Lipschitz regularization term being newly proposed.

The paper does not provide sufficient justification or analysis of why Lipschitz regularization is effective in preventing subcategorical memorization. The motivation currently appears intuitive rather than empirically grounded.

The authors should focus more on analyzing and justifying the contribution of the Lipschitz regularization term—e.g., through ablation studies, visualization of latent trajectories, or quantitative metrics.

Instead of positioning SURE as a standalone unlearning framework, the authors might consider integrating the Lipschitz regularization term into existing unlearning methods (such as MACE or ACE) to demonstrate broader utility.

Overall, the technical contribution in its current form seems insufficient for ICLR-level significance, as the novelty is limited and the empirical justification is incomplete

**Questions:**

-	How is the sample $x$ chosen for the Lipschitz regularization?
-	How does the proposed term interact with or affect other unlearning methods such as MACE, ACE or recent approaches when combined with them?
-	Can the authors provide a more detailed analysis of the effect of perturbation strength on both unlearning and generation quality?
-	How is the concept space $C$ defined and how sensitive is the performance to the choice of this space?

---

> ### Author Response · Authors · 2025-11-29
>
> **General comment:** We sincerely thank Reviewer $ \textcolor{blue}{PSKP} $ for their encouraging feedback on our work, as well as for their insightful comments and suggestions.
>
> - *Important problem:* We appreciate the recognition of the problem setting's importance, especially the challenge of subcategorical unlearning, which reviewers agree is both **novel** and **practically** relevant.
>
> - *Lipschitz regularization:* We also thank the reviewer for their positive comments regarding the motivation behind applying Lipschitz regularization to the latent space. As noted, utilizing Lipschitz continuity to **decrease category-level memorization** is both **interesting** and **intuitive.**
>
>
> We have now carefully addressed all of the reviewers concerns and questions here:
>
> **Response 1.1:**
>
> > W1: perceived novelty of the work.....
>
> We apologize for this concern. We want to highlight that our work addresses **category-level concept erasure** using a single parent concept, which we believe introduces a novel setting for concept erasure in text-to-image diffusion models.
>
> In particular, we extend the traditional $\mathcal{L}_1$ loss to enable the erasure of an **entire semantic category** using only the parent concept without requiring access to individual class-level concepts. This formulation is new, as previous L1-based erasure methods primarily works on single or multiple known concepts.
>
> Our $\mathcal{L}_3$ objective also differs from the adversarial concept loss used in [1]. Specifically, **we are the first to map related concepts to a neutral concept** and discover these related concepts by maximizing their divergence from the neutral concepts.
>
> This **joint formulation**, with our proposed Lipschitz regularization, is a novel approach to category-level concept erasure. Unlike existing methods that target individual or small groups of concepts, our approach erases an entire semantic category using only a single parent category concept, a formulation that, to the best of our knowledge, has not been previously explored in the literature. We have highlight this more in the revised version of the manuscript (Section 2 and 4).
>
> --------
>
> **Response 1.2:**
>
> >W2: why Lipschitz regularization is effective.....
>
> We apologize for this concern. Our motivation for using Lipschitz regularization is grounded in strong empirical evidence. By training the model to align its output for a target concept with its random perturbations, we can successfully reduce memorization of that target concept (i.e., forget). This also links with Feldman (2020) [2], which observes that generalized models are often forced to memorize certain information; our objective is to protect the broader generalization while removing the relevant memorized data. Our approach prevents over-reliance on memorized target concepts while preserving broader generative capability. Importantly, Lipschitz regularization reduces the model's sensitivity to small perturbations of concept variants. In category erasure, this is crucial, as the model must forget not only the explicit prompt (e.g., "gun") but also semantically close variants (e.g., "rifle," "SMG," "firearm").
>
> We also want to emphasize that we provide empirical ablations demonstrating that Lipschitz regularization is essential for preventing subcategorical memorization:
>
> - *Component-Wise result:* In Table 4 (left, appendix) we shows that removing the Lipschitz term severely weakens erasure performance: $\mathrm{Acc}_E$ increases from $10.00$ to $25.20$. This indicates that the Lipschitz constraint directly eliminates subcategorical detail, rather than merely altering overall model behaviour.
>
> - *Sensitivity to the number of perturbation samples ($n$):* Table 4 (right) reveals a clear pattern: moderate smoothing ($n=5$) produces the good category erasure. This aligns with the expected behavior of Lipschitz regularization, where too few samples lead to insufficient smoothing and too many samples lead to over-smoothing that impedes the necessary category-level shift.
>
> - *Sensitivity to perturbation strength ($\sigma$)*: In Table 7, we further show the category erasure in relation to perturbation strength. Specifically, moderate perturbations ($\sigma=0.10$) achieve the best trade-off between erasure and preservation.
>
> - *Overall empirical effectiveness:* In all three tasks,  object categories, explicit content removal, and artistic style erasure, our method achieves the highest harmonic mean scores (see Tables 1–3). These results demonstrate the significance of Lipschitz regularization in enabling SURE subcategorical erasure performance, which surpasses that of existing methods.
>
>
> Together, these ablations provide strong empirical justification for the role of Lipschitz regularization in preventing subcategorical memorization. Following the reviewer’s suggestion, we have including additional ablations of Lipschitz reg. on Blade Weapons, Musical Instruments, and Toys in Sec. B.1 and Table 5 (Appendx).

---

> ### Author Response · Authors · 2025-11-29
>
> ------
>
> **Response 1.3:**
>
> >W3: justifying the contribution of the Lipschitz regularization term through ablation studies....
>
> We apologize for this concern. As noted in our Response 1.2, **Lipschitz regularization plays a central role in enabling subcategorical forgetting**. Following the reviewer’s valuable suggestion, we have strengthened both the theoretical and empirical justification of this component.
>
> Specifically, we have added (i) an **theoretical explanation** of why Lipschitz smoothing suppresses memorized subcategory-level structure (Section A.6), and (ii) additional **empirical analyses**, including ablations on **perturbation strength (Section B.5), number of perturbation samples (Section B.2), and per-component removal studies (Section B.1)**. These are now included and more highlighted in the revised manuscript in Section B and A (Appendix). Together, these analyses provide additional clarity on how the Lipschitz term contributes to category-level unlearning.
>
> -----
>
> **Response 1.4:**
>
> >W4: Integrating the Lipschitz regularization term into existing unlearning methods........
>
> We thank the reviewer for this valuable suggestion. We agree that Lipschitz regularization can serve as a plug-and-play. Because the regularization operates on the intermediate latent outputs of the diffusion model, it is independent of the specific architectural choices made by methods such as MACE or ACE. Following the reviewer’s suggestions, we have now incorporated the Lipschitz regularization term into the ACE baseline (on guns dataset) and report the results in Table 1 (rebuttal) and Table 11 of the revised manuscript. The performance improvements obtained in this setting indicate that the **Lipschitz can be used as plug-and-play.**
>
>
> **Table: 1** Effect of adding Lipschitz regularization to the ACE baseline on the gun dataset.
> | **Method**   | **AccE (↓)** | **AccL (↑)** |
> |--------------|--------------|--------------|
> | ACE          | 76.00        | 70.22        |
> | ACE + Lip    | 41.00        | 68.70        |
>
>
>
> --------

---

> ### Author Response · Authors · 2025-11-29
>
> ------
>
> **Response 1.5:**
>
> >W5: Overall, the technical contribution....
>
> We sincerely apologize for the reviewer concern regarding the technical contribution. We would like to clarify that the revised manuscript significantly enhances both the novelty and empirical justification of SURE. The core contributions of our framework are summarized below:
>
> - **New Problem Setting of Category Erasure from a Single parent Concept:** We propose new setting where the goal is to erase an entire semantic category (e.g., guns, toys, artistic styles) using only a single category concept. Prior works primarily focus on individual concept removal or small, curated sets; To our knowledge, no prior work proposes a scalable formulation capable of removing an entire hypernym and its hyponyms using only a single parent-category prompt. This problem formulation itself is a key contribution and expands the scope of unlearning in diffusion models.
>
>
> - **Related Concept Discovery and Erasure:** We use a differentiable mechanism to identify semantically related concepts and explicitly map them to a neutral concept. This enables the model to generalize forgetting beyond the given parent prompt, a capability not present in existing methods.
>
>
> - **Introduction of Lipschitz regularization for concept forgetting:** To the best of our knowledge, we are the first to present Lipschitz regularization within the concept erasure literature of text-to-image diffusion models. The revised version now includes an expanded theoretical explanation and extensive empirical support for the role of Lipschitz smoothing in erasing subcategorical memorization.
>
>
> - **Extensive and Diverse Evaluation:** We evaluate our method on **three distinct tasks: object removal, explicit content suppression, and artistic style erasure** and demonstrating that our approach generalizes across different concept types and domains. We also show effectiveness of our framework with strong performance gains across all tasks (Tables 1–3).
>
>
> Following the reviewer’s feedback, we have added/improved several new analyses:
> 1. component ablations isolating the Lipschitz term,
> 2. sensitivity to perturbation count ($n$),
> 3. sensitivity to perturbation strength ($\sigma$),
> 4. integration of the Lipschitz term into the ACE baseline (Table 11), showing plug-and-play generalization.
>
>
>
> We believe that our work holds significant potential. As this is a **new setting**, we appreciate all the review questions and the opportunity to address them. Since we propose a novel setting, it is understandable that comparisons with prior works may not be immediately straightforward when encountering something new; it is natural to relate it to familiar approaches. However, we were happy to clarify during the discussion.
>
> ------
>
> **Response 1.6:**
>
> > Questions 1: How is the sample $x$ chosen for the Lipschitz regularization?
>
> We thank the reviewer for the question. The sample $x$ used in the Lipschitz regularization is obtained directly from the  erased model conditioned on the parent category embedding $c_e$. We first generate an image sample $x = D\left(z_0\right)$ where $z_0$ is the final latent obtained from the diffusion trajectory and $D(\cdot)$ is the VAE decoder.  This image represents the model’s current expression of the target category. After obtaining $x$, we apply Gaussian perturbations $\xi \sim \mathcal{N}(0,\sigma^2 I)$ to produce $x' = x + \xi$, and both $x$ and $x'$ are encoded by the VAE encoder $f_{\theta'}(\cdot)$ for computing the Lipschitz loss (Eq. 8).
>
> ------
>
> **Response 1.7:**
>
> > Questions 2: How does the proposed term interact ....
>
> We appreciate the reviewer's question.  As discussed in Response 1.4, the Lipschitz term operates solely on the latent outputs of the diffusion model and is therefore architecture and method agnostic. As shown in Table 1 (rebuttal), adding our term to ACE (ACE+Lip) improves erasure performance, demonstrating that it can be used as a plug-and-play for existing unlearning frameworks.
>
> ------

---

> ### Author Response · Authors · 2025-11-29
>
> **Response 1.8:**
>
> > Questions 3:  effect of perturbation strength.....
>
>  We thank the reviewer for this question.  As shown in Table 8 of the revised manuscript, the behavior is non-monotonic: very small perturbations ($\sigma = 0.01$) produce weak unlearning ($\mathrm{Acc}_E = 20.00$), since the perturbations are too small to meaningfully smooth the model’s latent response around the erased category.
>
> Moderate perturbation strength ($\sigma = 0.10$) yields the good forgetting performance ($\mathrm{Acc}_E = 10.00$) while maintaining high locality ($\mathrm{Acc}_L = 67.20$). This range provides enough variation to suppress sub categorical memorization. We observe no degradation in visual quality on non-target concepts at this setting (illustrated in Appendix C).
>
> For larger perturbations ($\sigma = 0.30$), forgetting remains effective but is slightly weaker than at the optimal setting, and generation quality also bit decline. This reflects an over-regularization effect: strong perturbations induce excessive smoothing, which interferes with the model’s ability to move toward the neutral concept and also introduces mild decrease in the generation of non-target content. We have highlighted this behavior more clearly in the revised version of the manuscript (Appendix Section B.5).
>
>
> ------
>
> **Response 1.9:**
>
> > Questions 4:  How is the concept space  defined .....
>
> We thank the reviewer for this question. The concept space $C$ is constructed from the CLIP text vocabulary, which provides a broad and semantically rich embedding space. As detailed in Appendix A.4, we select the 100 CLIP tokens. This filtered set forms the concept space $C$ used for related-concept discovery.
>
> Regarding sensitivity, our method is not highly dependent on the precise choice of $C$ because the objective
>
> $
> \max_{c_r \in C \setminus E}
> \| \epsilon_{\theta'}(\tau(c_r)) - \epsilon_\theta(\tau(c_n)) \|_2^2
> $
>
> automatically identifies the most influential related concepts within the chosen space. In practice, varying the size of $C$ (e.g., 100 or 200 candidates)  for initial pruning results in negligible changes in performance. This robustness arises because the differentiable Gumbel Softmax operator consistently selects only a small number of high-impact related concepts regardless of the initial pool (see Appendix B.3 and Table 6). To further validate this, we added experiments in the revised manuscript showing that erasure performance remains stable across different choices of $|C|$ (Appendix B.10). For example, increasing the pool to 200 CLIP tokens results in a minor GPU memory overhead of approximately 20 MB, while maintaining similar performance, with accuracy values of $\mathrm{Acc}_E = 10.40$ and $\mathrm{Acc}_L = 67.80$. Further increasing the pool to 500 CLIP tokens yields accuracy values of $\mathrm{Acc}_E = 10.00$ and $\mathrm{Acc}_L = 68.00$.
>
> Thus, while $C$ defines the search domain, the optimization process itself determines which concepts influence unlearning.
>
> -----
>
> We trust that our responses effectively address all the reviewers concerns.
>
> [1] Erasing Undesirable Concepts in Diffusion Models with Adversarial Preservation.
>
> [2] Does learning require memorization? a short tale about a long tail

---

### Author Response · Authors · 2025-11-30

**Author Remarks and Summary:** We wish to convey our heartfelt gratitude to the Area Chairs and Reviewers for generously investing their time and providing encouraging feedback on our work.

**Important problem, Novelty, Lipschitz regularization, Subcategorical erasure as a meaningful, Experimental Results and Performance Gains**

----

- Our work addresses the novel problem of category erasure. We believe this introduces a novel approach to concept erasure in text-to-image diffusion models. SURE enables the model to automatically erase entire subcategories such as SMG, revolver, pistol, rifle, and shotgun using only a single guns prompt. This ability is critical for real-world scalability, especially in scenarios where related concepts are unknown. Our approach integrates this setting with Lipschitz regularization, forming a joint formulation that, to the best of our knowledge, has not been previously explored in the literature. Unlike prior works that focus on a specific, defined concept, SURE generalizes erasure across multiple subcategories while preserving unrelated concepts.

- We believe that our work holds significant potential. As this is a new setting, we appreciate all the review questions and the opportunity to address them. Most of the points addressed in the rebuttal expand upon content already present in the paper, and all additional experiments and reviewer concerns have been incorporated into the revised manuscript.


**The major highlights of the revisions include:**

- Expanded motivation (Sections 2 and 4).
- Theoretical justification of the Lipschitz term (Appendix A.6).
- Detailed analysis of the contribution of Lipschitz regularization (Appendix B.1; Tables 4 and 5).
- Plug-and-play integration with prior methods (Table 11).
- Additional suggested experiments (Appendix B.8, B.9, B.10, B.11).
- Analysis of unintended erasure and preservation controllability (Appendix B.8, B.11).
- Justification of key design decisions for $\mathcal{L}_{3}$ (Appendix B.3; Tables 6 and 7).


----

We believe that we have adequately addressed all concerns raised by the reviewers, and all suggestions have been incorporated into either the main paper or the Appendix. As per the updated guidelines, we did not receive follow-up responses from the reviewers. We are concerned that the manuscript may be unfairly penalized due to a misunderstanding and the lack of post-rebuttal engagement. We hope that the clarifications and additional results provided during the rebuttal will contribute positively to the final assessment.

---

### Meta-Review · Area_Chair_WcA9 · 2026-01-07

**Summary:**

This paper proposes SURE (Subcategorical Unlearning via Regularized Erasure), a method for erasing entire semantic subcategories from text-to-image diffusion models using only a single parent-category prompt (e.g., "gun" to remove "pistol" "rifle", etc.). The approach combines an adversarial subcategory discovery mechanism over a CLIP-based concept space with Lipschitz regularization in the latent space to promote smooth responses around the target category. Reviewers generally acknowledged that the problem of subcategorical erasure is timely and practically relevant for safety alignment in generative models, and that the empirical results, particularly the strong erasure performance while preserving unrelated generation capabilities, are compelling. However, several fundamental concerns were raised:

- **(1) Theoretical and mechanistic justification for Lipschitz regularization:** While the paper shows that the Lipschitz term improves erasure performance, it does not establish a causal or formal link between latent-space smoothness and the erasure of subcategorical knowledge. The motivation remains largely empirical and intuitive.
- **(2) Reliance on CLIP-based semantic proximity for subcategory discovery:** The method assumes that subcategories reside near the parent concept in CLIP embedding space, but semantic proximity does not necessarily imply a hyponymic (category–member) relationship. This conflation of lexical similarity with conceptual subsumption raises concerns about the reliability and completeness of the discovery process.
- **(3) Evaluation within a closed concept set:** All experiments evaluate erasure on pre-defined subcategory lists drawn from the same CLIP vocabulary used during training. While standard in the field, this setup does not fully probe the method's behavior when CLIP neighborhoods include semantically adjacent but non-subcategorical concepts.
- **(4) Scope of problem formulation:** The "single parent prompt" setting assumes a clear hierarchical relationship between the prompt and its subcategories, which may not hold for abstract or safety-critical concepts where category boundaries are fuzzy.

**Reviewer Concerns:**

Following the rebuttal, I believe several technical and empirical concerns have been adequately addressed. The authors provided extensive new ablations (e.g., on perturbation number, Lipschitz weight, Gumbel-Softmax variants), clarified implementation details (e.g., VAE freezing, gradient flow), and demonstrated that the Lipschitz regularizer can be effectively plugged into prior methods like ACE to improve erasure. Concerns about over-erasure, hyperparameter sensitivity, and experimental completeness were largely resolved through additional results in the rebuttal.

However, two core issues remain insufficiently resolved. First, despite empirical evidence that the Lipschitz term is effective, the rebuttal does not offer a mechanistic explanation for how it enables generalization from a parent concept to unseen subcategories. Second, the method's dependence on CLIP cosine similarity as a proxy for subcategorical structure is not critically examined. While the authors show robustness to the size of the concept set $C$, they do not analyze whether the adversarial search actually recovers true hyponyms or is misled by semantically close but unrelated terms. Since CLIP similarity reflects co-occurrence rather than logical category membership, this assumption lacks strong justification and may fail in nuanced safety scenarios.

**Reviewer Scores:**

Based on the rebuttal, I reckon that Reviewer QZ8U and Reviewer TvHz raised substantive but addressable concerns; their scores (both originally 4) would likely have improved had they seen the full rebuttal, though not necessarily to acceptance. In contrast, Reviewer PSKP and Reviewer jzz5 maintained deeper skepticism about the method and theoretical foundation. Even with the rebuttal, I believe they would have retained their initial reject scores (both originally 2).

---

### Decision · Program_Chairs · 2026-01-26

Reject